# Quantitative X-ray Maps Analaysis of Composition and Microstructure of Permian High-Temperature Relicts in Acidic Rocks from the Sesia-Lanzo Zone Eclogitic Continental Crust, Western Alps

**Michele Zucali** [1,*,†] , **Luca Corti** [1,†] , **Manuel Roda** [1,†] , **Gaetano Ortolano** [2,†] , **Roberto Visalli** [2,†] and **Davide Zanoni** [1,†]

1   Dipartimento di Scienze della Terra "A. Desio", Università degli Studi di Milano, Via Mangiagalli 34, 20133 Milano, Italy; itroc11@gmail.com (L.C.); manuel.roda@unimi.it (M.R.); davide.zanoni@unimi.it (D.Z.)
2   Dipartimento di Scienze Biologiche, Geologiche e Ambientali, Università degli Studi di Catania, Corso Italia, 57, 95129 Catania, Italy; ortolano@unict.it (G.O.); rvisalli@unict.it (R.V.)
*   Correspondence: michele.zucali@unimi.it; Tel.: +39-02-503-15547
†   These authors contributed equally to this work.

**Abstract:** Three samples of meta-acidic rocks with pre-Alpine metamorphic relicts from the Sesia-Lanzo Zone eclogitic continental crust were investigated using stepwise controlled elemental maps by means of the Quantitative X-ray Maps Analyzer (Q-XRMA). Samples were chosen with the aim of analysing the reacting zones along the boundaries between the pre-Alpine and Alpine mineral phases, which developed in low chemically reactive systems. The quantitative data treatment of the X-ray images was based on a former multivariate statistical analytical stage followed by a sequential phase and sub-phase classification and permitted to isolate and to quantitatively investigate the local paragenetic equilibria. The parageneses thus observed were interpreted as related to the pre-Alpine metamorphic or magmatic stages as well as to local Alpine re-equilibrations. On the basis of electron microprobe analysis, specific compositional ranges were defined in micro-domains of the relict and new paragenetic equilibria. In this way calibrated compositional maps were obtained and used to contour different types of reacting boundaries between adjacent solid solution phases. The pre-Alpine and Alpine mineral parageneses thus obtained allowed to perform geothermobarometry on a statistically meaningful and reliable dataset. In general, metamorphic temperatures cluster at 600–700 °C and 450–550 °C, with lower temperatures referred to a retrograde metamorphic re-equilibration. In all the cases described, pre-Alpine parageneses were overprinted by an Alpine metamorphic mineral assemblage. Pressure-temperature estimates of the Alpine stage averagely range between 420 to 550 °C and 12 to 16.5 kbar. The PT constraints permitted to better define the pre-Alpine metamorphic scenario of the western Austroalpine sectors, as well as to better understand the influence of the pre-Alpine metamorphic inheritance on the forthcoming Alpine tectonic evolution.

**Keywords:** pre-Alpine; mineral relicts; X-ray image analysis; Sesia-Lanzo Zone; geothermobarometry

## 1. Introduction

The recognition of metamorphic relicts followed by the identification of the thermodynamic equilibria affecting basements rocks is of paramount importance to quantitatively constrain the tectonics and geodynamics of continental crust at extensional as at convergent margins (e.g., [1–5]). Relicts are often highly scattered preventing the reconstruction of a complete record of superimposed cycles [6]. This scattering is largely caused by deformation and metamoprhism partitioning. In highly sheared domains, detailed structural and metamorphic studies are often unfeasible as intense development of metamorphic reaction progresses occur [7–9], and frequently prevent the preservation of sufficiently large relict volumes.

A successful approach to this problem was mapping strain partitioning within different rock types, allowing a partial reconstruction of the superimposed mechanical and metamorphic incomplete re-equilibration stages [10,11]. The procedure involves firstly to separate Coronite-Tectonite-Mylonite domains (CTM approach) at field scale [12]. In general, the largest amount of textural and mineralogical relicts is found in coronite domains. However, the CTM approach may prove unsuccessful where relicts are too small to be mapped. This limitation, well known in most of the pre-Alpine relicts metamorphic complexes of western Alps, is normally overcome by means of a detailed local analysis that can be performed combining microstructural analysis with X-ray elemental maps, obtained by Electron Microprobe Analyzer (EMPA). This already routinely performed approach was recently reinforced using increasingly efficient processing tools aimed at extracting numerical petrological constraints [5,13–19].

A good example of reconstruction of the tectonometamorphic evolution from structural and metamorphic relicts is the Sesia-Lanzo Zone (SLZ), where its pre-Alpine evolution was reconstructed starting from few sparse relicts (Table 1 and references therein). These relicts are mainly preserved in metabasites [12,20] and minor metapelites or metacidic rocks [21].

The pre-Alpine evolution of the Sesia-Lanzo Zone was ascribed to the post-Variscan extension that produced, during Permian time, large high temperature/migmatitic terranes associated with magma sourced from the migmatitic crust or from continental mantle [22–27]. Such lithosphere-scale extension was the precursor of the rifting and oceanization that started during the Triassic to Jurassic [26–28] that led to the Tethys Ocean. According to the literature, the SLZ inherited several Permian lithospheric-scale heterogeneites that were activated during the Alpine evolution.

In this contribution we will study acidic samples to select pre-Alpine mineral and structural remnants to detail pre-Alpine relicts in less chemically reactive systems and to better define such heterogeneities at the SLZ scale. We sampled CTM textures to test the approach against different strain states. We selected three localities, two in the EMC and one in the RCTU: Lago della Vecchia (LdV), Monte Mucrone (MM), and Rocca Canavese (RCT). The three localities were chosen where previous studies described metre-scale relicts of pre-Alpine protoliths and fabrics associated with well-preserved metamorphic minerals [9,19,21,29,30], but no detailed evolution and pressure-temperature estimates were reported. The three localities differ in lithologic and texture types, where LdV are metaintrusives preserved in Coronite Alpine fabric, MM—high temperature metapelites wrapped in Tectonite Alpine fabric, RCT are metapelite relicts within a Mylonite Alpine domain.

Following the workflow described in [14,18], we will apply a combined microstructural and minerochemical analysis to access the mineral relicts (e.g., [1,4,18]). Furthermore, we will investigate the potential mineral zoning pattern within the single-phase and/or alongside the active boundaries. The results will be discussed relating the mineral compositions with the degree of fabric development. In this contribution we briefly describe the workflow (Section 3) used for the minerochemical analysis applied to the collected samples (Section 4). Finally, we present the petrological implications of the analysis (Section 5) and we discuss the results in light of the tectonic evolution of the Sesia-Lanzo Zone and the rock memory during tectonic cycles (Section 6).

**Table 1.** Sample code, geographical location, tectonic unit, lithology, pre-Alpine mineral assemblage, PT estimate, ages (where present), and references for the Permian metamorphic samples and metaintrusives in the Sesia-Lanzo Zone. Apc = Austroalpine permian crust; Apg = Austroalpine permian gabbros; Apgr = Austroalpine permian metagranitoids. Mineral abbreviations after [31]. Coordinate system WGS84-UTM 32N.

| Code | Location | Coord_X | Coord_Y | Complex | Group | Lithology | Assemblage | Temp (°C) | Pres (GPa) | Age (Ma) | References |
|---|---|---|---|---|---|---|---|---|---|---|---|
| Apc1a | Rassa Valley | 421,099.247 | 5,064,527.1 | EMC | Continental crust | Basic granulites | Opx, Pl, Grt, Qz, Amp | 725 ± 75 | 0.8 ± 0.1 | 270 ± 25 | [12] |
| Apc1b | Alpe Maccagno | 411,076.47 | 5,063,883.44 | EMC | Continental crust | Acid granulites | Sil, Bt, Crd, Pl, Qz | 700 ± 50 | 0.7 ± 0.1 | 270 ± 25 | [12] |
| Apc21 | Val del Lys | 409,620.187 | 5,054,520.75 | EMC | Continental crust | Metapelites | Grt, Mnz, Zrn | 750 ± 50 | 0.8 ± 0.1 | 289 ± 7.5 | [12,32] |
| Apc22 | Laghetto Monte Rosso | 416,915.996 | 5,053,677.63 | EMC | Continental crust | Metapelites | Zrn | 750 ± 50 | 0.8 ± 0.1 | 297 ± 16 | [12,32] |
| Apc31 | Rechantier, Val del Lys | 408,509.324 | 5,051,572.75 | EMC | Continental crust | Metapelites | Grt | 750 ± 50 | 0.8 | 270 ± 25 | [17] |
| Apc32 | Lillianes, Val del Lys | 409,670.997 | 5,054,025.17 | EMC | Continental crust | Metapelites | Grt | 900 ± 40 | 0.65 | 270 ± 25 | [17] |
| Apc33 | Faye, Val del Lys | 406,650.646 | 5,053,947.73 | EMC | Continental crust | Metapelites | Grt | 780 ± 20 | 0.8 | 270 ± 25 | [17] |
| Apc34 | Liévanere, Val del Lys | 406,315.051 | 5,052,450.46 | EMC | Continental crust | Metapelites | Grt | 730 ± 60 | 0.6 | 270 ± 25 | [17] |
| Apc35 | Monte Soglio | 383,905.469 | 5,023,811.2 | EMC | Continental crust | Metapelites | Grt, Aln, Zrn | | | 292 ± 11 | [33] |
| Apc36 | Chiusella | 400,789.256 | 5,044,935.93 | EMC | Continental crust | Metapelites | Grt, Aln, Zrn | | | 279 ± 3.6 | [33] |
| Apc37 | Monte Mucrone | 417,406.317 | 5,053,631.22 | EMC | Continental crust | Metapelites | Grt, Aln, Zrn | | | 286 ± 2.9 | [33] |
| Apc1c | Tesso Valley | 383,260.963 | 5,024,757.83 | EMC | Continental crust | Acid granulites | Sil, Bt, Pl, Qz | 725 ± 75 | 0.8 ± 0.1 | 270 ± 25 | [12] |
| Apc1c | | | | | | Basic granulites | Opx, Pl, Grt, Qz, Amp | | | | |
| Apc1d | Tesso Valley | 384,318.412 | 5,025,493.44 | EMC | Continental crust | Acid granulites | Sil, Bt, Pl, Qz | 725 ± 75 | 0.8 ± 0.1 | 270 ± 25 | [12] |
| Apc1d | | | | | | Basic granulites | Opx, Pl, Grt, Qz, Amp | | | | |
| Apc1g | Monte Mucrone | 418,662.517 | 5,053,171.02 | EMC | Continental crust | Acid granulites | Sil, Bt, Pl, Qz | 725 ± 75 | 0.8 ± 0.1 | 270 ± 25 | [12] |
| Apc1g | | | | | | Basic granulites | Opx, Pl, Grt, Qz, Amp | | | | |
| Apc1h | Monte Cossarello | 418,202.757 | 5,067,607.5 | EMC | Continental crust | Acid granulites | Sil, Bt, Pl, Qz | 725 ± 75 | 0.8 ± 0.1 | 270 ± 25 | [12] |
| Apc1h | | | | | | Basic granulites | Opx, Pl, Grt, Qz, Amp | | | | |
| Apc1i | Rassa Valley | 424,179.642 | 5,068,573 | EMC | Continental crust | Acid granulites | Sil, Bt, Pl, Qz | 725 ± 75 | 0.8 ± 0.1 | 270 ± 25 | [12] |
| Apc1i | | | | | | Basic granulites | Opx, Pl, Grt, Qz, Amp | | | | |
| Apc1l | Plaida Lake | 412,363.799 | 5,071,055.7 | EMC | Continental crust | Acid granulites | Sil, Bt, Pl, Qz | 725 ± 75 | 0.8 ± 0.1 | 270 ± 25 | [12] |
| Apc39 | Carema | 405,862.248 | 5,046,976.88 | EMC | Continental crust | Marbles | Ttn | | | 270 ± 25 | [34] |
| Apc1l | | | | | | Basic granulites | Opx, Pl, Grt, Qz, Amp | | | | |
| Apc1e | Verres | 398,295.13 | 5,055,423.85 | GM | Continental crust | Acid granulites | Sil, Bt, Pl, Qz | 725 ± 75 | 0.8 ± 0.1 | 270 ± 25 | [12] |
| Apc1e | | | | | | Basic granulites | Opx, Pl, Grt, Qz, Amp | | | | |
| Apc1f | Verres | 399,536.483 | 5,057,814.6 | GM | Continental crust | Acid granulites | Sil, Bt, Pl, Qz | 725 ± 75 | 0.8 ± 0.1 | 270 ± 25 | [12] |
| Apc1f | | | | | | Basic granulites | Opx, Pl, Grt, Qz, Amp | | | | |
| Apc2 | Gressonay | 412,885.125 | 5,069,313.67 | II DK | Continental crust | Acid granulites | Sil, Bt, Pl, Qz | 700 ± 50 | 0.65 ± 0.05 | 270 ± 25 | [35–37] |
| Apc2 | | | | | | Basic granulites | Opx, Pl, Grt, Qz, Amp | | | | |
| Apc23 | A. Piana, Val Mastallone | 431,900.305 | 5,085,354.12 | II DK | Continental crust | Metapelites | Grt, Qz, Pl, Kfs, Sil, Bt, Zrn, Rt, Ilm, Ap | 756 ± 84 | 0.7 ± 0.1 | 279.4 ± 3.4 | [32,36,38] |
| Apc24 | Val d'Egua | 426,741.174 | 5,083,852.94 | II DK | Continental crust | Metapelites | Qz, Grt, Rt, Zrn | 800 ± 29 | 0.7 ± 0.1 | 279 ± 9 | [32,36,38] |
| Apc26 | Val Sesia | 419,113.845 | 5,075,964.95 | II DK | Continental crust | Metapelites | Qz, Wm, Grt, Pl, Kfs, Bt, Zrn, Ilm | 623.5 ± 42.5 | 0.7 ± 0.1 | 295.5 ± 9.5 | [32,36,38] |
| Apc27 | Val del Lys | 410,438.392 | 5,065,029.94 | II DK | Continental crust | Metapelites | Qz, Wm, Grt, Pl, Kfs, Bt, Zrn, Mnz | 676 ± 69 | 0.7 ± 0.1 | 283.5 ± 8.5 | [32,36,38] |
| Apc28 | Valle di Ribordone | 386,185.126 | 5,033,008.01 | II DK | Continental crust | Metapelites | Qz, Grt, Pl, Rt, Zrn, Ilm | 794 ± 94 | 0.7 ± 0.1 | 271.7 ± 4.4 | [32,36,38] |
| Apc29 | Val Soana | 387,888.606 | 5,034,930.75 | II DK | Continental crust | Metapelites | Qz, Pl, Grt, Bt, Rt, Zrn, Ilm, Ap | 673.5 ± 46.5 | 0.7 ± 0.1 | 290 ± 15 | [32,36,38] |
| Apc30 | S. Maria, Val Mastallone | 433,518.584 | 5,083,622.93 | II DK | Continental crust | Metapelites | Grt, Qz, Bt, Pl, Kfs, Sil, Rt, Ilm, Zrn, Mnz | 756 ± 84 | 0.7 ± 0.1 | 252 ± 35 | [32,36,38] |
| Apc38 | Case Fremt | 387,447.135 | 5,019,838.48 | RCTU | Continental crust | Metapelites | Grt, Bt, Wm | | | 284 ± 11 | [19] |
| Apg1 | Corio-Monastero | 381,575.286 | 5,018,201.72 | EMC | Continental gabbro | Gabbro-norite | Cpx, Opx, Pl, Amp, Ilm, Ap | 850 ± 70 | 0.75 ± 0.15 | 270 ± 25 | [20] |
| Apg2 | Val Sermenza | 427,828.893 | 5,077,828.99 | GM | Continental gabbro | Gabbro | Pl, Amp, Cpx, Mag, Zo ± Ms ± Chl | | | 288 ± 4 | [39] |
| Apgr1 | Lago della Vecchia | 416,868.975 | 5,059,627.23 | EMC | Continental crust | Metagranites | Bt, Kfs, Wm, Aln, Pl, Ttn | 710 ± 19 | 0.52 ± 0.21 | 270 ± 25 | [21] |
| Apb1 | Monte Mars | 417,286.743 | 5,052,353.70 | EMC | Continental crust | Metabasics | Ti-rich Amph | 720 ± 48 | 0.3 ± 0.05 | 270 ± 25 | [29] |

## 2. Geological Setting

The Sesia-Lanzo Zone (SLZ) is the largest portion of Austroalpine eclogitic continental crust that was subducted and exhumed during on-going oceanic subduction during the Alpine convergence [29,40–53]. The SLZ is bounded by rocks of the Penninic domain to the northwest and separated by the Southern Alps by the westernmost part of the Periadriatic Lineament (PL) to the southeast, here named the Canavese System (Figure 1). The SLZ consists of micaschists, paragneisses, and orthogneisses, with minor marble and metabasite [40]. The SLZ is divided into four complexes (Figure 2a,b; [40,54]): the II Dioritic-Kinzigitic Zone (IIDK), the Gneiss Minuti Complex (GMC), the Eclogitic Micaschists Complex (EMC), and the Rocca Canavese Thrust Sheets Unit (RCTU). The IIDK preserves a pervasive pre-Alpine high-temperature (HT) metamorphic imprint. The GMC and the EMC display a dominant greenschist and eclogitic Alpine metamorphic imprint, respectively [9,11,55–59]. The Rocca Canavese Thrust Sheets Unit (RCTU) is characterised by a dominant blueschist facies Alpine imprint with no eclogite facies relics [11,19,60–63].

Eclogite facies metamorphism developed at conditions of 13–20 kbar and 500–600 °C [12,29,40,59,64,65] at 85–65 Ma [53,66–68]. Blueschist facies metamorphism developed at ca. 60 Ma [3,46,49,69–73] at pressures lower than 15 kbar and temperatures ranging between 450 and 550 °C [3,29,53]. Finally, the greenschist facies re-equilibration [46] anticipated the Periadriatic intrusions emplaced shallower than 10 km [74–76].

Within the western Austroalpine, domain the Dent Blanche system and the IIDK, in SLZ, preserve dominant structural, petrographic, and metamorphic tracers of the Permian magmatism and high temperature metamorphism (Figure 2). IIDK and Valpelline Series (Dent Blanche) both record Permian granulite facies metamorphism that is associated with penetrative fabrics and folds and intrusion of pegmatite dykes. In particular, IIDK consists of lower crustal metapelites and metabasics and better than the other SLZ units preserves pre-Alpine relics that largely escaped Alpine eclogitic metamorphism [35,36,38,77], but only overprinted by a gentle Alpine blueschist facies metamorphism in coronitic or localised high-strain domains [54]. In the metapelite of IIDK, the typical pre-Alpine assemblage is Qz, Grt, Pl, Bt, Wm (mineral abbreviations after [31]). Kfs is also common together with Rt, Ilm, Ap, Zrn, and Mnz. Sil is locally preserved (Table 1). Grt is usually Fe- and Mg-rich, and Bt has a high Ti content. In the metabasites, the typical pre-Alpine assemblage is Opx, Pl, Grt, Qz, Amp with Fe- and Mg-rich Grt. The pre-Alpine metamorphic imprint was characterised by low P/T gradients (Figure 2b) and it is recorded under amphibolite to granulite facies conditions (Figure 2c). Recent U/Pb analyses on zircons indicate a Permian age for the pre-Alpine metamorphism in the IIDK (Table 1).

In addition to IIDK, pre-Alpine geologic remnants are documented also in the EMC and RCTU (Figure 2a). Although in EMC the eclogitic Alpine metamorphism is dominant, tracers of pre-Alpine metamorphism and magmatism are preserved as relict kinzigites, granites, and gabbro in microscopic up to a few square kilometres domains. Such relics are preserved in variably deformed and transformed domains [2,9,21,30,40,47,49,77,78]. Given the strong Alpine imprint in the EMC, the pre-Alpine textures and metamorphic assemblages are better preserved in the rocks volumes that were poorly deformed during the Alpine convergence (e.g., [9]). Thus, pre-Alpine features were mostly documented in rocks with Alpine coronite textures (Figure 2a, Table 1). The pre-Alpine assemblages in basic and acidic granulites consist of Opx, Pl, Grt, Qz, Amp, and Bt and Sil, Bt, Pl, Grt, Crd, Kfs, and Qz, respectively [12]. In the tectonitic and mylonitic domains, the pre-Alpine metamorphism is recorded by few relics, such as Grt, Aln, Mnz, and Zrn [12,33,64]. The pre-Alpine Grt occurs as a pluri-millimetre-sized fractured porphyroclast sealed by Alpine Grt rims [8,33,64]. The Grt core is Fe- and Mg-rich and the rim is Ca-rich [8,12,13,64]. P-T estimates on acidic and basic coronite granulites agree with those performed on the pre-Alpine Grt in mylonitic rocks and indicate a re-equilibration under granulite facies conditions [12,64]. Permian ages of these pre-Alpine imprints in EMC were obtained from Zrn and Aln [32,33,64]. In the EMC several Carboniferous gabbros and Permian granites are variably affected by Alpine deformation and metamorphism (e.g., Monte Mucrone,

Monte Mars, Lago della Vecchia; Figure 2a, Table 1) and, in the least deformed types (coronite types), igneous texture and mineral relicts can be detected [20,21,29,39,45,79]. Furthermore, in the Corio and Monastero metagabbros, a pre-Alpine metamorphic stage under amphibolite facies conditions is interpreted as the effect of exhumation after the emplacement occurred at ca. 0.6–0.8 GPa and 800–900 °C [20]. Shallower emplacement conditions have been estimated for the metagranites of Lago della Vecchia (Table 1) at ca. 0.4–0.6 GPa and 700 °C [21].

In GMC, metapelites and acidic and basic granulites (Figure 2 and Table 1) preserved high-temperature Permian mineral assemblages that are overprinted by Alpine coronitic assemblages developed under greenschist to eclogite facies conditions [12,80]. In RCTU only microscale relicts were reported, such as porphyroclasts of white mica and garnet in garnet-bearing gneiss and K-feldspar and biotite porphyroclasts in orthogneiss [19]. As in the EMC, Grt occurs as a pluri-millimetre-sized porphyroclast surrounded by foliation, with a fractured Fe and Mg-rich core and Alpine Ca-rich rims [19]. Age is not available for this assemblage, but the mineral composition is strikingly similar to that of the Permian parageneses recorded in other portions of the SLZ [2,3,8,12,19,33,64,70,72,81].

Despite the strong Alpine re-equilibration of the SLZ under greenschist to eclogite facies conditions, several Permian metamorphic and magmatic relicts were detected (Figure 2a). Within coronite textures clear pre-Alpine mineral assemblages are preserved, while within tectonite to mylonite textures only few Grt relicts attest the pre-Alpine metamorphism. The pressure-temperature (PT) estimates on the pre-Alpine assemblages indicate a granulite to amphibolite facies re-equilibration under a high thermal regime during the Permian, accompanied by the intrusion of gabbros and granitoids (Figure 2). This HT event was ascribed to the Permian-Triassic lithospheric thinning that affected Pangea and consequent asthenospheric upwelling [22–27].

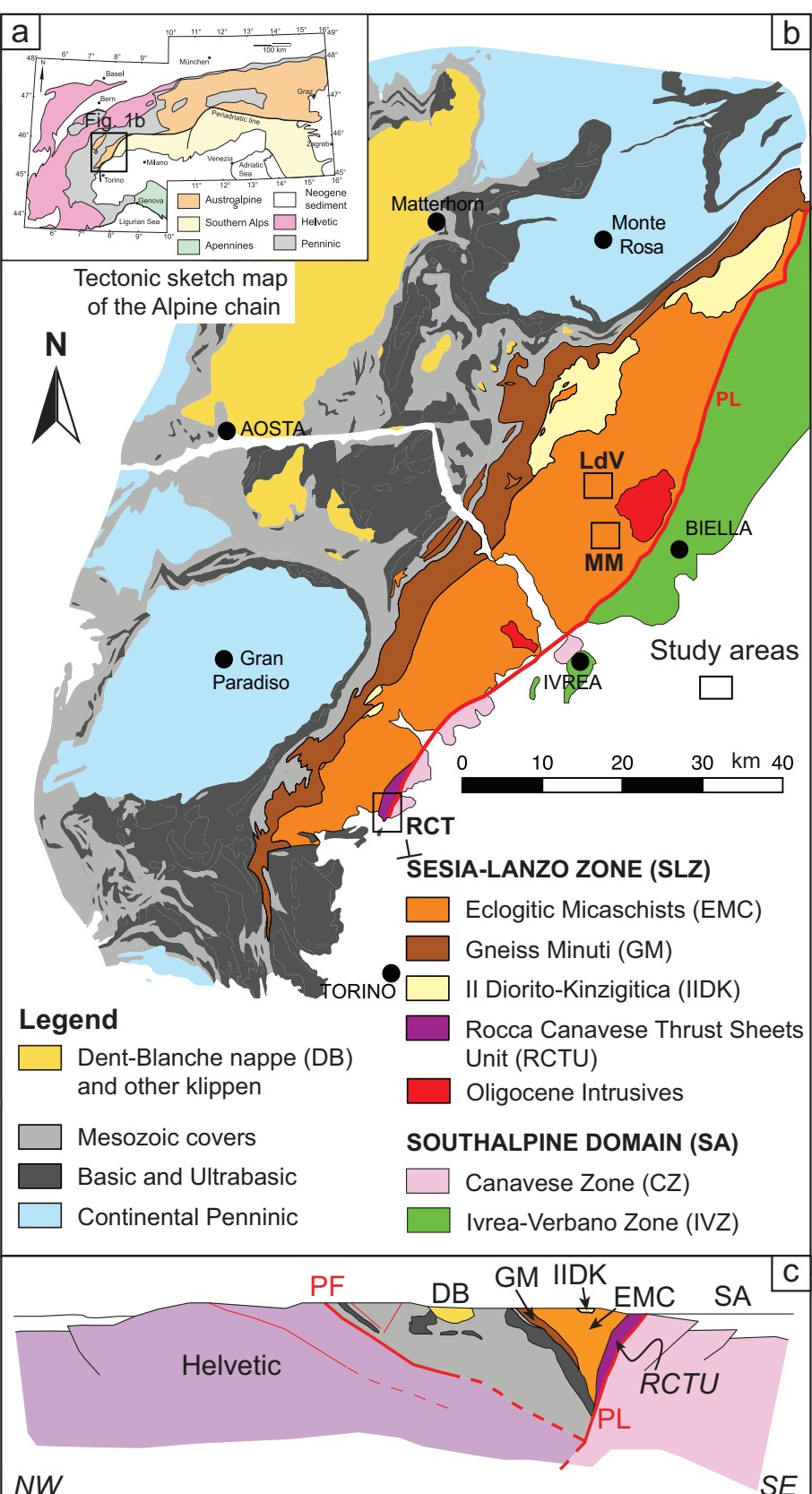

**Figure 1.** (**a**) Simplified sketch map of the tectonic domains of the Alps; (**b**) Sketch map of the Sesia-Lanzo Zone (modified after [19,40,55,74,78,82,83]) with the location of sampling areas: Lago della Vecchia (LdV), Monte Mucrone (MM), and Rocca Canavese (RCT); (**c**) simplified geological cross-section through the Western Alps modified after [84]. PL = Periadriatic lineament; PF = Penninic front.

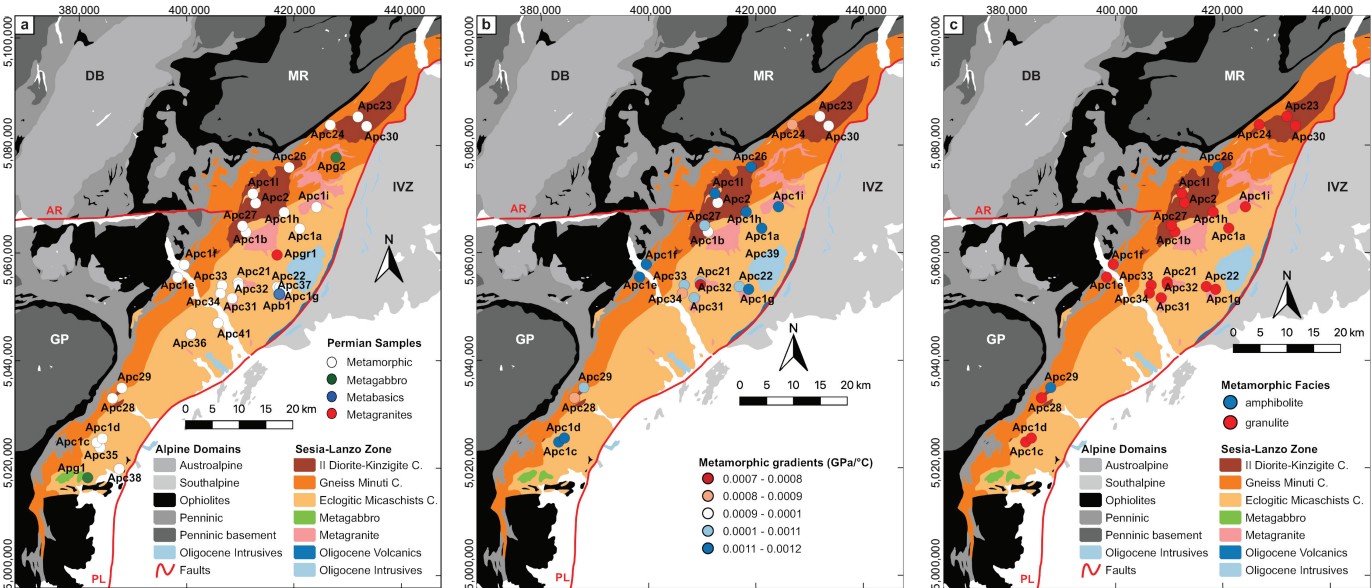

**Figure 2.** (**a**) Location of Permian metamorphic samples (white dots) and metagabbros (green dots) in the Sesia-Lanzo Zone. (**b**) Metamorphic states (P/T ratio) obtained from Permian metamorphic samples in the Sesia-Lanzo Zone. (**c**) Metamorphic facies conditions of Permian metamorphic samples in the Sesia-Lanzo Zone. Sample code, geographical location, tectonic unit, lithology, pre-Alpine mineral assemblage, PT estimate, ages (where present), and references are in Table 1. DB = Dent-Blanche nappe; GP = Gran Paradiso Massif; MR = Monte Rosa Massif, IVZ = Ivrea-Verbano Zone; PL = Periadriatic lineament; AR = Aosta-Ranzola fault. Figure modified after [85]. Coordinate system WGS84-UTM 32N.

## 3. Analytical Method

Mineral chemical analyses and X-ray maps of the selected rock thin sections were elaborated by means of the Quantitative X-Ray Map Analyzer tool (Q-XRMA) developed by [18]. This method is useful to quantitatively extrapolate the sequence of the metamorphic assemblages related to the different fabrics (e.g., [5,19,86]), as well as to investigate the potential mineral zoning pattern within a single mineral phase and/or alongside the border of two mineral phases. Q-XRMA is an image processing tool package based on several image analysis functions written in Python and largely based on the ArcGIS® library functions, in the same line of several tools progressively developed to address different geosciences-related issues (e.g., [5,14–16,86–91]).

Q-XRMA is used to classify rock-forming minerals, starting from an initial stage that uses as input of the process an array of low-resolution major elements X-ray maps (i.e., Al, Ca, Fe, K, Mg, Mn, Na, Si, Ti) at thin section scale (acquired with a dwell time of 160 ms at resolution ranging from $350 \times 250$ to $650 \times 350$ with an average pixel size of 60 µm), to pass successively at a high-resolution array of wavelength-dispersive spectroscopy (WDS) X-ray maps at microstructural domain scale, acquired with a higher dwell time ranging from 130 ms to 160 ms at a resolution ranging from $450 \times 350$ to $770 \times 600$ for pixel sizes spanning from 5 to 2 µm. This last analytical stage can be very well-performing when it comes investigating the mineral zoning patterns within a single or between two phases as well as to calibrating the maps for pixel-based chemical analysis and end-member component maps, by using spot chemical analyses as internal standards (e.g., [13,18,86]).

For calibration and comparison, mineral spot analyses for calibration and comparison were determined using a Jeol, JXA-8200 electron microprobe (WDS, accelerating voltage of 15 kV, beam current of 15 nA), operating at the Department of Earth Sciences, University of Milano. Natural silicates were used as standards and the results were processed for matrix effects using a conventional ZAF procedure [92,93]. Mineral formulae were calculated on the basis of 12 oxygens for garnet, 6 for pyroxene, 23 for amphibole, 22 for mica and biotite, 12.5 for epidote, 8 for feldspar, and 20 for titanite. $Fe^{3+}$ was recalculated based on charge

balance [94]. The classification of amphiboles follows IMA 2012 recommendations [95] and we used a spreadsheet proposed by [96]. The classification of pyroxenes is after [97].

More specifically, the Q-XRMA procedure is divided into three different cycles (Figure 3): (i) the first cycle is useful to classify mineral phases at thin section and microdomain scale as well as to infer the associated modal percentage by a multivariate statistical data handling of the X-ray maps through the Principal Components Analysis (PCA) and the supervised Maximum Likelihood Classification (MLC; [87]; (ii) the second cycle performs a deeper analysis of selected mineral phases to detect mineral zoning and to calibrate X-ray maps thanks to a series of image analysis functions based on a multiple linear regression analysis [18]. As a result, the element concentration values can be calculated for each pixel related to each mineral phase investigated; (iii) the third cycle allows to manage the calibrated X-ray maps to obtain the maps of end-member components, as well as to quantify the chemical variations within each mineral phase. For each calculated pixel, the stoichiometry agrees with the structural formula of corresponding mineral species.

This method allows defining a refined analysis of the metamorphic assemblages that characterize the different fabrics. In this work, this method was applied to investigate the three pre-Alpine rock samples that differently recorded the eclogitic metamorphism of the Sesia-Lanzo continental crust.

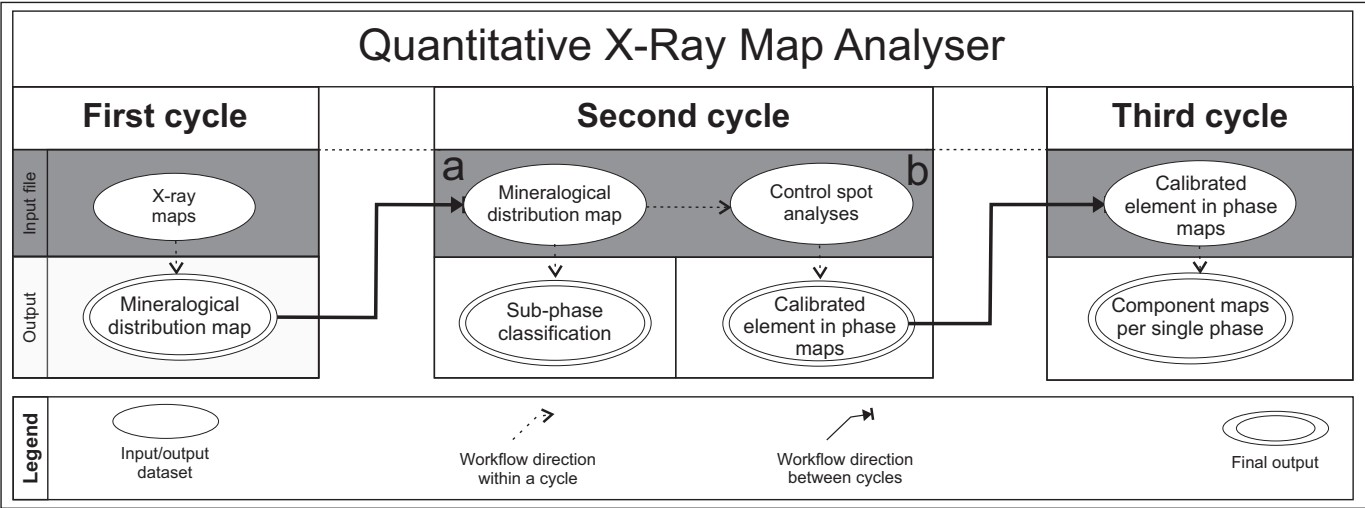

**Figure 3.** Simplified flow chart of the Quantitative X-ray Map Analyzer workflow (modified after [18]: First cycle: classification of the main recognized mineral phase. Second cycle: (**a**) potential sub-phase classification; (**b**) pixel calibration in a.p.f.u. Third cycle: the transformation of calibrated a.p.f.u. maps into end member maps for solid solution mineral phase.

## 4. Samples Description and Data Analysis

Here we present data obtained from three samples collected in three different localities of the Sesia-Lanzo Zone that namely are Lago della Vecchia (LdV) and Monte Mucrone (MM), within EMC, and Rocca Canavese (RCT), within the RCTU (Figure 4). Results obtained through the image-assisted analysis performed via Q-XRMA are shown in the following sections for each of the three samples, which contain pre-Alpine mineral relicts within mm- to μm-scale domains.

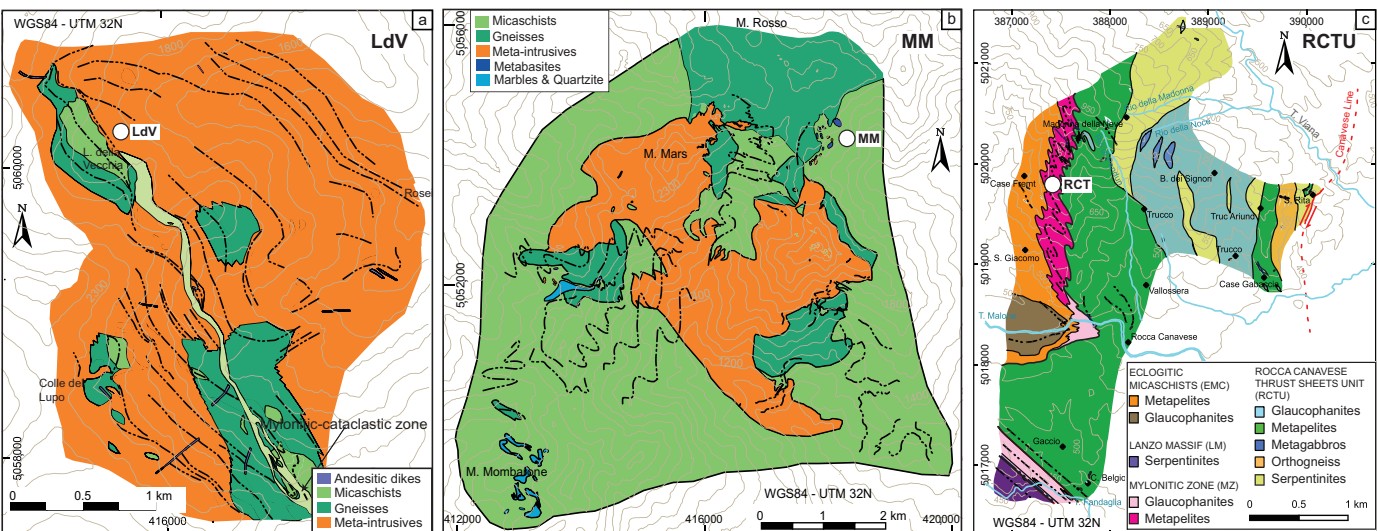

**Figure 4.** Geological maps of the (**a**) Lago della Vecchia (LdV) area modified after [30], (**b**) Monte Mucrone (MM) area modified after [9], and (**c**) Rocca Canavese Thrusts Unit (RCTU) area modified after [19,63]. White dots represent the locations of analysed samples. Black dashed lines represent the trajectory of the dominant Alpine foliation.

### 4.1. Lago Della Vecchia Metagranites

At LdV metagranitoids, micaschists, and banded gneisses outcrop. Even if this zone is within the EMC, blueschist facies mineral assemblages mark the pervasive foliation in these rocks [30]. The Lago della Vecchia metagranites are characterised by metre-sized coronitic domains where relicts of the pre-Alpine texture are partly preserved (Figure 4a) and display the original igneous texture and minerals (Figure 5a,b). K-feldspar, biotite, quartz, and white mica are still visible to the naked eye, whereas pristine Aln is visible under the microscope [21]. The coronitic domains are scattered in tectonitic to mylonitic metagranites, where eclogite or blueschists facies minerals replace igneous textures and minerals during the Alpine subduction [30]. At km-scale, the tectonitic-mylonitic bands run NW-SE, with a thickness varying from few meters to tens of meters. The field-foliation is marked by phengitic mica, quartz, epidote, and blue amphibole. At the microscale, albitic plagioclase also marks the main foliation [21]. Within this foliation, centimetre-sized igneous relicts can be recognized.

The First Cycle of the Q-XRMA procedure (Figure 3) allows recognising and separating main mineral phases and defining their distribution in the sample (Figure 6a,b). The segmented mineral map obtained permitted to define the relative modal abundances of each mineralogical phase used to define a qualitative bulk-rock composition (Table 2; Figure 6b). Segmented mineral map permitted also to highlight specific features not easily visible at the common optical microscopy analysis. Biotite occurs as a single generation of large grains, always rimmed by garnet (black in Figure 6b). White mica may occur in two generations, the first one as large single grains, the second one as fine-grained aggregates that rim the first generation of white mica or partly replace plagioclase (ex-plagioclase domains in Figure 6b). Epidote (yellow in Figure 6b) systematically forms corona texture between ex-plagioclase domains and white mica aggregates. A new generation of plagioclase (pink) occurs as large domains rimming K-feldspar.

Igneous biotite ($Bt_I$) and white mica ($Wm_I$) (Figure 7a,b) are frequently rimmed by garnet ($Grt_I$), epidote, and a second generation of biotite ($Bt_{II}$) and white mica ($Wm_{II}$). The most interesting feature is a corona around biotite grains (Figure 7a,b) where an almost continuous garnet ($Grt_I$) corona reproduces the original grain boundary. Even more interesting, $Grt_I$ is zoned, showing a darker side against biotite and a lighter one toward the ex-plagioclase domain [65]. This last feature is better highlighted by the sub-phase classification results of the Q-XRMA second cycle (Figure 3) shown in Figure 7b. In Figure 7a, the former classification step of the first cycle permitted to depict: (a) a local

generation of amphibole with plagioclase within the garnet rim; (b) ex-plagioclase domains replaced by aggregates of white mica, epidote, plagioclase, and amphibole.

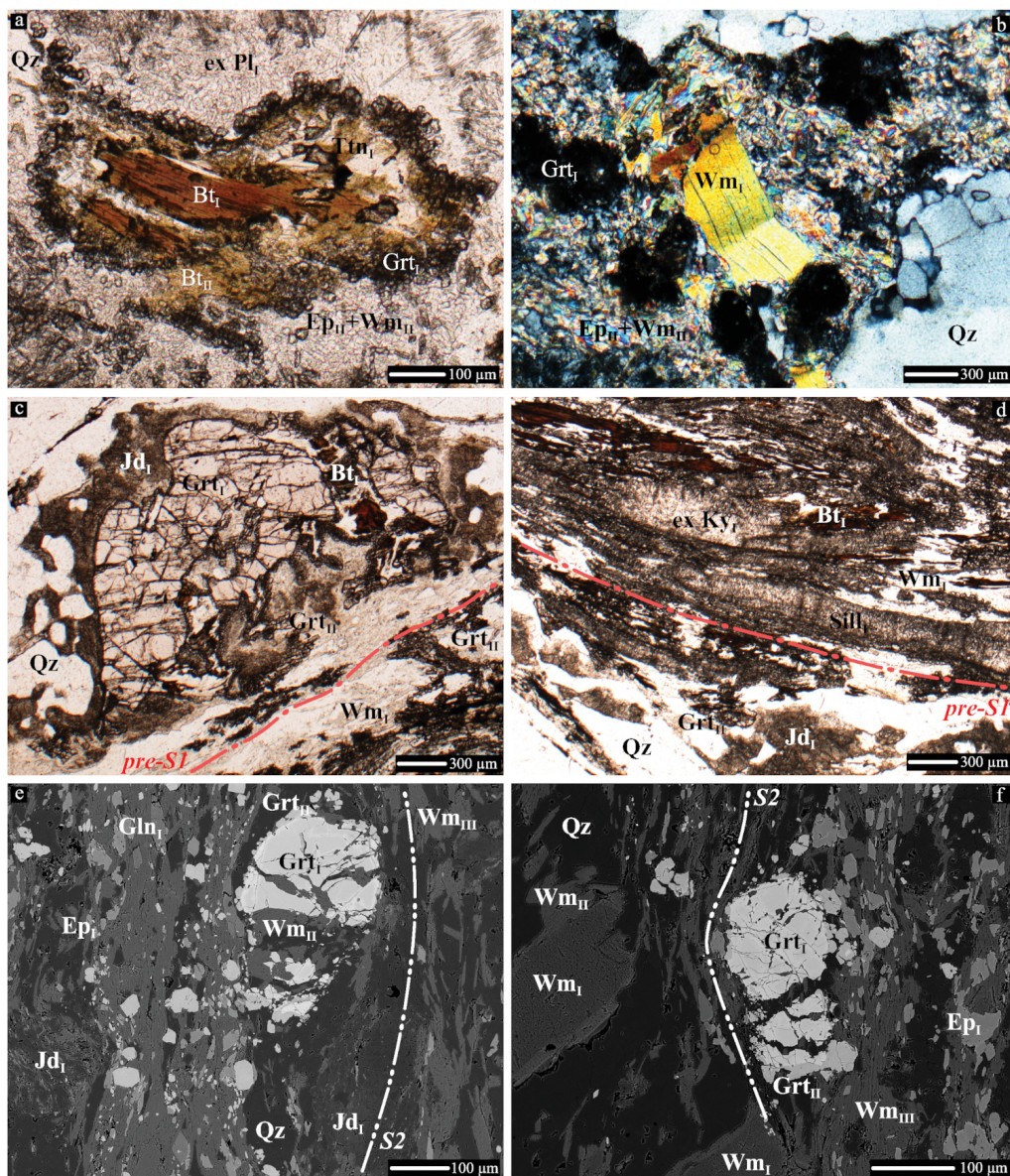

**Figure 5.** Detailed mineralogical assemblages at the microdomain scale for LdV, MM, and RCT investigated samples: (**a**,**b**) LdV micrographs of garnet corona (Grt$_I$) defining the primary boundaries between igneous biotite (Bt$_I$) and ex-PlI domain (**a**-plane polarised light) and rimming relict white mica (**b**-crossed polars); (**c**,**d**) MM micrographs of microgarnet (Grt$_{II}$) and JdI grown around Grt$_I$ (**c**-plane polarised light) and pre-Alpine Sil$_I$ and Pl microdomains (**d**-plane polarised light); (**e**,**f**) RCT back-scattered electron images displaying Grt$_I$ rimmed by Grt$_{II}$ and S2 foliation marked by Gln$_I$, Ep$_I$, Jd$_I$, and fine grained Wm$_{III}$.

The Q-XRMA Second Cycle results (Figure 7b) allowed detecting a peculiar zoning pattern in every single garnet grain of the corona rimming the pre-Alpine biotite and, more interestingly, a third generation of garnet (Figure 7b), two biotite generations, and three white mica compositional types (Figure 7b). In details Grt$_{Ic}$ and Grt$_{Ib}$ correspond to the two generations visible by optical microscopy, while Grt$_{Ia}$ was only revealed by this analysis. Similarly, the first generation of biotite is most likely of igneous origin. The second generation of biotite is probably related to a prograde stage pre-dating the eclogitic metamorphism because it shows straight boundaries with Grt$_{Ib}$, Wm, and igneous

plagioclase-domains. The first generation of white mica ($Wm_I$) is in contact with $Bt_I$ and rimmed by $Grt_I$, whereas new generations of $Wm_{II}$ and $Wm_{III}$ grew in contact with $Bt_{II}$ and epidote (Figure 7b), respectively.

The third cycle of the Q-XRMA (Figure 3) permitted to transform each mineral phase pixel, when correctly classified, into a specific atom per formula unit (a.p.f.u.) value, stoichiometrically consistent with the phase classified. The calibrated images thus obtained permitted to relate the mineral-chemical variation within a single phase to different texture types. Moreover, Fe and Ca contents in garnet are clearly correlated with the microdomain chemistry (Figure 8a,b). Higher Fe contents in garnet occur close to biotite, while high Ca contents are related to the ex-plagioclase microdomains. Biotite composition (Figure 8a,b) is also strongly related to the igneous microdomain. Ti content is higher in the core of the large biotite grains, suggesting its close relation with igneous composition ($Bt_I$). Ti content decreases, approaching to 0, as biotite get closer to the garnet rim ($Bt_{II}$). Fe + Mg + Mn content shows a similar trend (Figure 8b). Additionally, $Wm_I$ shows composition variations related to texture. Al-poor (Figure 9a) and high $X_{Mg}$ content (Figure 9b) occur where white mica is in contact with igneous biotite. $Wm_{II}$ and $Wm_{III}$ present an increase in Al (Figure 9a) and lower content of $X_{Mg}$ (Figure 9b).

**Table 2.** Average compositions of minerals from Lago della Vecchia obtained via Q-XRMA. All values are calculated on the base of 605 calibrated pixels for $Grt_{Ia}$; 1412 calibrated pixels for $Grt_{Ib}$ and 7454 calibrated pixels for $Grt_{Ic}$; 8006 calibrated pixels for $Bt_I$ and 6782 calibrated pixels for $Bt_{II}$; 735 calibrated pixels for $Wm_I$; 8169 calibrated pixels for $Wm_{II}$ and 15,118 calibrated pixels for $Wm_{III}$. Alm = Fe/(Fe + Mg + Ca + Mn), Grs = Ca/(Fe + Mg + Ca + Mn), Prp = Mg/(Fe + Mg + Ca + Mn), Sps = Fe/(Fe + Mg + Ca + Mn).

| Mineral | Garnet | | | Biotite | | White Mica | | |
|---|---|---|---|---|---|---|---|---|
| **Sub-Phase** | **$Grt_{Ia}$** | **$Grt_{Ib}$** | **$Grt_{Ic}$** | **$Bt_I$** | **$Bt_{II}$** | **$Wm_I$** | **$Wm_{II}$** | **$Wm_{III}$** |
| Al | 2.03 | 2.03 | 2.02 | 2.65 | 2.85 | 4.14 | 4.38 | 4.66 |
| Ca | 1.18 | 0.99 | 1.33 | 0.00 | 0.00 | 0.01 | 0.01 | 0.02 |
| Fe | 1.51 | 1.63 | 1.44 | 2.83 | 2.85 | 0.33 | 0.33 | 0.34 |
| K | 0.00 | 0.00 | 0.00 | 1.82 | 1.84 | 1.90 | 1.91 | 1.91 |
| Mg | 0.12 | 0.19 | 0.10 | 2.02 | 2.02 | 0.66 | 0.55 | 0.49 |
| Mn | 0.06 | 0.04 | 0.04 | 0.03 | 0.04 | 0.00 | 0.00 | 0.01 |
| Na | 0.00 | 0.00 | 0.00 | 0.00 | 0.00 | 0.02 | 0.03 | 0.06 |
| Si | 3.03 | 3.04 | 3.02 | 5.84 | 5.79 | 6.74 | 6.69 | 6.57 |
| Ti | 0.00 | 0.00 | 0.00 | 0.26 | 0.15 | 0.18 | 0.10 | 0.02 |
| Cations | 7.92 | 7.92 | 7.94 | 15.45 | 15.53 | 13.98 | 14.01 | 14.07 |
| Alm | 52.74 | 57.26 | 49.59 | | | | | |
| Grs | 40.97 | 34.48 | 45.90 | | | | | |
| Prp | 4.22 | 6.76 | 3.32 | | | | | |
| Sps | 2.07 | 1.50 | 1.23 | | | | | |
| $X_{Fe}$ | | | | 0.58 | 0.58 | 0.34 | 0.40 | 0.44 |
| $Al_{IV}$ | | | | 2.16 | 2.21 | | | |
| $Al_{VI}$ | | | | 0.48 | 0.64 | | | |

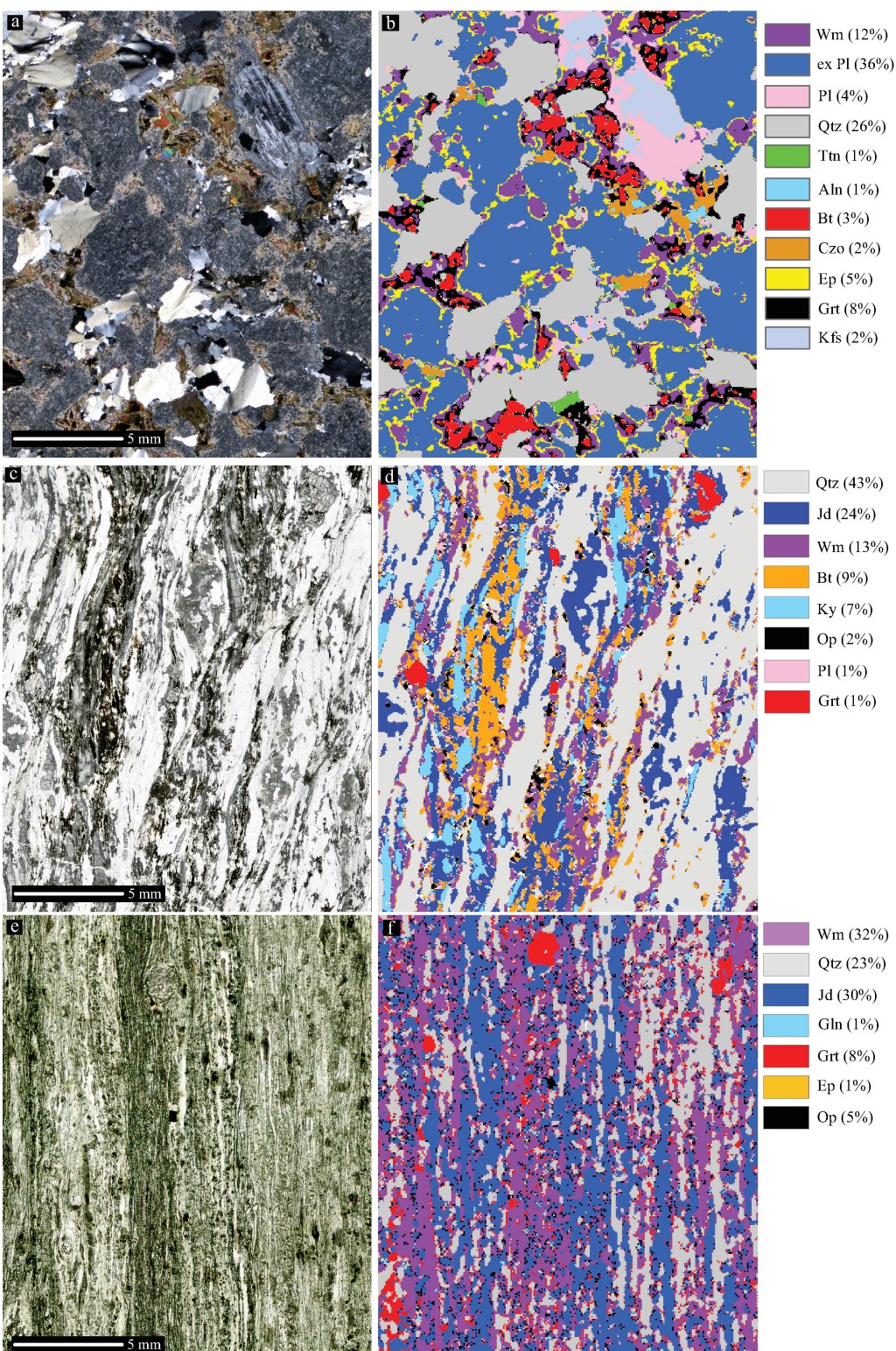

**Figure 6.** Investigated samples from LdV, MM, and RCT localities at the thin section scale: (**a**,**c**,**e**) optical thin section scans of LdV metagranite (crossed polars), MM metapelite (plane polarised light), and RCT mylonitic paragneiss (plane polarised light); (**b**,**d**,**f**) LdV, MM, and RCT mineral distribution maps and relative modal abundances obtained through the first cycle of the Q-XRMA.

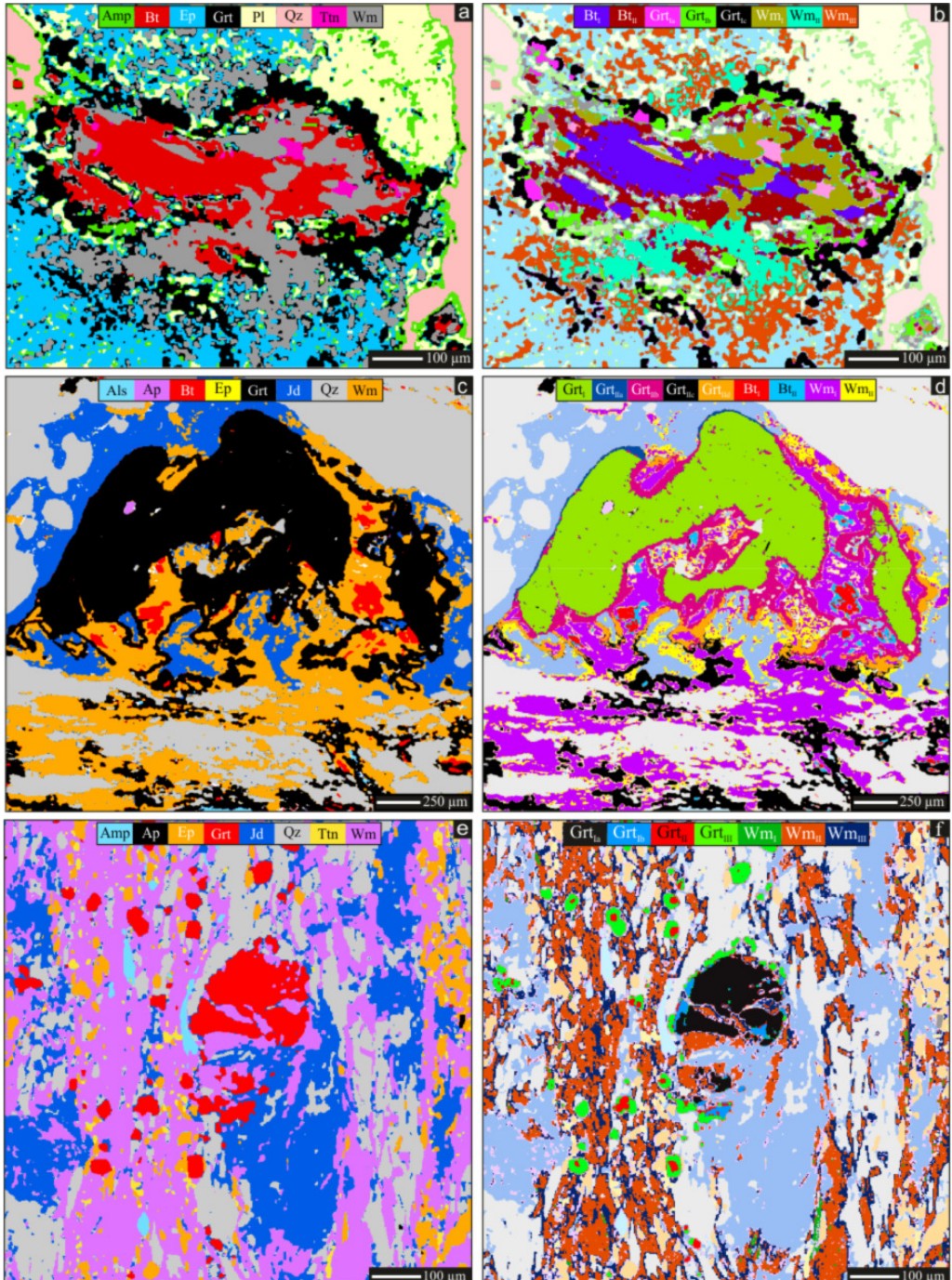

**Figure 7.** Results obtained with the Q-XRMA classification procedure: (**a**,**b**) LdV mineral map (**a**) obtained by the first cycle and sub-phase classification of Grt, Bt, and Wm (**b**) obtained by the second cycle; (**c**,**d**) MM mineral map (**c**) obtained by the first cycle and sub-phase classification of Grt, Bt, and Wm (**d**) obtained by the second cycle; (**e**,**f**) RCT mineral map (**e**) obtained by the first cycle and sub-phase classification of Grt, and Wm (**f**) obtained by the second cycle.

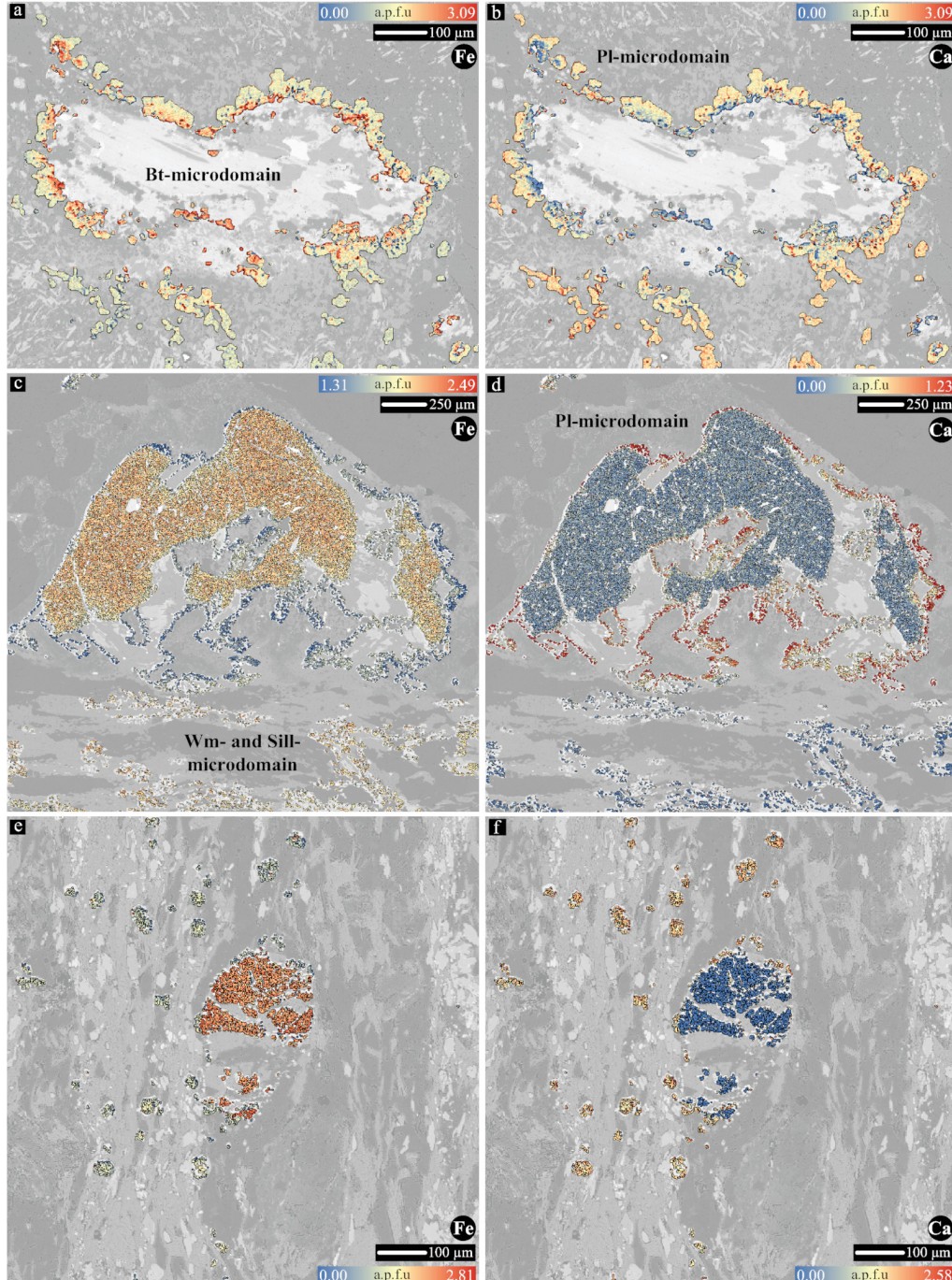

**Figure 8.** Calibrated maps of garnet: (**a**,**b**) calibrated maps of Fe and Ca of LdV garnets; (**c**,**d**) calibrated maps of Fe and Ca of MM garnets; (**e**,**f**) calibrated maps of Fe and Ca of RCT garnets.

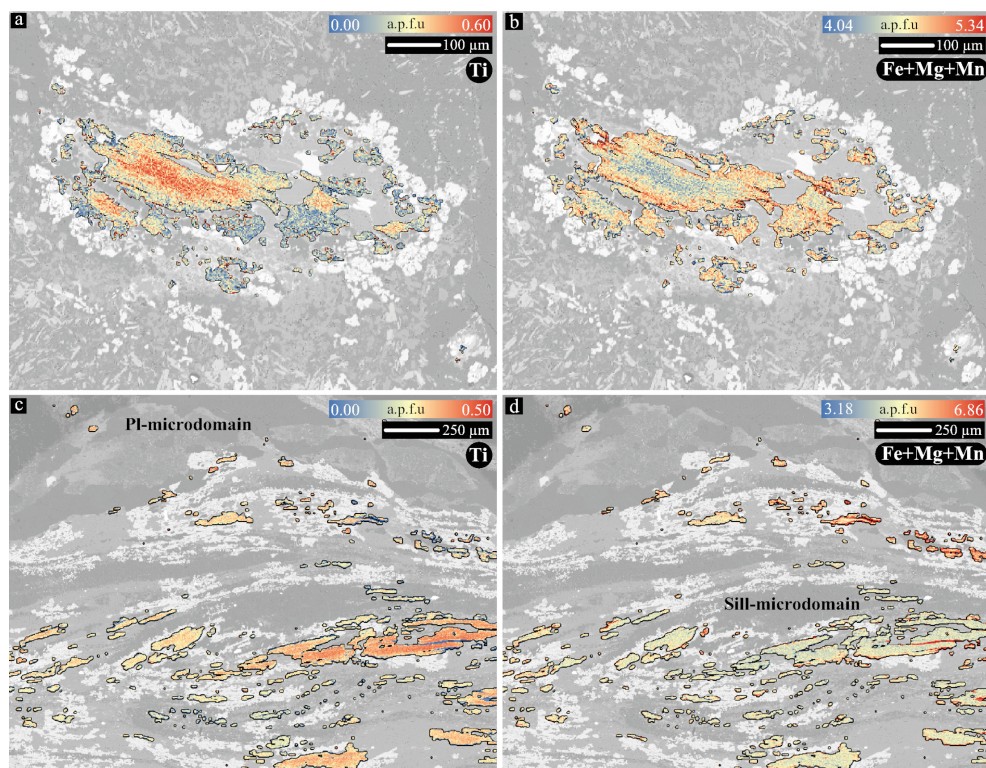

**Figure 9.** Calibrated maps of biotite: (**a**,**b**) calibrated maps of Ti and Fe + Mg + Mn of LdV biotite; (**c**,**d**) calibrated maps of Ti and Fe + Mg + Mn of MM biotite.

### 4.2. Monte Mucrone Metapelites

In the MM area, metagranitoids, micaschists, gneisses, and minor marble and quartzite outcrops. Locally metagranitoids display primary contact with metasedimentary rocks [49], which preserve a pre-Alpine metamorphism [44]. In particular, Permian intrusives outcrop [9] where eclogitic mineral assemblages are widespread in both coronitic to mylonitic domains (Figure 4). Country rocks are completely re-equilibrated under eclogite facies conditions but include meter-sized lenses of metapelite protoliths with Permian fabric and, locally, mineral phases [67]. Biotite-bearing gneiss occurs with mm-thick lithons defined by quartz, feldspar, and allumosilicate, which was formerly sillimanite.

At the micro scale, sillimanite domains are replaced by fine aggregates of kyanite, while plagioclase is replaced by omphacite. Biotite is largely replaced by phengitic white mica and large garnet porphyroblasts are rimmed by a second generation of garnet, which shows straight boundaries with omphacite and phengite grains (Figure 5c,d). Such relations are well described in the mineral maps (Figure 6d), where colour codes help to recognise the relict foliation (pre-S1) marked by biotite ($Bt_I$) as well as continuous domains composed of kyanite or jadeite replacing former sillimanite and plagioclase, respectively. Relative modal abundances of each mineralogical phase are presented in Table 3.

**Table 3.** Average compositions of minerals from Monte Mucrone obtained via Q-XRMA. All values are calculated on the base of 63,404 calibrated pixels for Grt$_I$; 879 calibrated pixels for Grt$_{IIa}$; 14,265 calibrated pixels for Grt$_{IIb}$; 41,557 calibrated pixels for Grt$_{IIc}$ and 5085 calibrated pixels for Grt$_{IId}$; 1694 calibrated pixels for Bt$_I$ and 1478 calibrated pixels for Bt$_{II}$; 49,492 calibrated pixels for Wm$_I$ and 41,644 calibrated pixels for Wm$_{II}$.

| Mineral | Garnet | | | | | Biotite | | White Mica | |
|---|---|---|---|---|---|---|---|---|---|
| Sub-Phase | Grt$_I$ | Grt$_{IIa}$ | Grt$_{IIb}$ | Grt$_{IIc}$ | Grt$_{IId}$ | Bt$_I$ | Bt$_{II}$ | Wm$_I$ | Wm$_{II}$ |
| Al | 1.99 | 1.99 | 2.00 | 2.02 | 2.00 | 2.69 | 2.65 | 4.88 | 4.37 |
| Ca | 0.10 | 0.99 | 0.14 | 0.09 | 0.44 | 0.00 | 0.00 | 0.00 | 0.00 |
| Fe | 2.30 | 1.75 | 2.24 | 2.26 | 2.06 | 2.16 | 2.51 | 0.19 | 0.25 |
| K | 0.00 | 0.00 | 0.00 | 0.00 | 0.00 | 1.81 | 1.77 | 1.94 | 1.95 |
| Mg | 0.37 | 0.17 | 0.51 | 0.64 | 0.43 | 2.67 | 2.46 | 0.41 | 0.57 |
| Mn | 0.23 | 0.10 | 0.12 | 0.02 | 0.06 | 0.01 | 0.01 | 0.00 | 0.00 |
| Na | 0.00 | 0.00 | 0.00 | 0.00 | 0.00 | 0.00 | 0.00 | 0.05 | 0.04 |
| Si | 3.00 | 3.00 | 2.99 | 2.99 | 3.00 | 5.78 | 5.80 | 6.40 | 6.63 |
| Ti | 0.00 | 0.00 | 0.00 | 0.00 | 0.00 | 0.32 | 0.26 | 0.13 | 0.17 |
| Cations | 8.00 | 8.00 | 8.00 | 8.00 | 7.99 | 15.44 | 15.47 | 14.01 | 13.99 |
| Alm | 76.69 | 58.25 | 74.52 | 75.31 | 68.64 | | | | |
| Grs | 3.26 | 32.93 | 4.72 | 2.85 | 14.77 | | | | |
| Prp | 12.28 | 5.55 | 16.85 | 21.27 | 14.51 | | | | |
| Sps | 7.76 | 3.28 | 3.91 | 0.58 | 2.08 | | | | |
| X$_{Fe}$ | | | | | | 0.45 | 0.50 | 0.32 | 0.30 |
| Al$_{IV}$ | | | | | | 2.22 | 2.20 | | |
| Al$_{VI}$ | | | | | | 0.47 | 0.46 | | |

Grt (red in Figure 6d) occurs as isolated porphyroblasts. Close up to garnet and sillimanite microdomains (Figure 6c,d) shows well preserved garnet (GrtI) rimmed by fine-grained garnet (Grt$_{II}$) and jadeite (Jd$_I$). A few biotite grains also occur within this microdomain. Relict microlithons reveal the prismatic sillimanite microdomains replaced by kyanite. White mica grains also occur locally (Wm$_I$).

The sub-phase classification via Q-XRMA first cycle shows a detailed mineral map (Figure 7c) where the microdomain confirms the distribution of garnet either as large porphyroblasts or as fine layers within the white mica microdomain or in association with jadeite. Q-XRMA second Cycle (Figure 7d) shows different generations of garnet, biotite, and white mica. Grt$_I$ represents the core of the large porphyroblasts, most likely preserving the original Permian composition. Grt$_{IIa}$, Grt$_{IIb}$ represent the rim of Grt$_I$ in contact with Jd (ex-plagioclase) or white mica, respectively. Grt$_{IIc}$ mainly occurs along the foliation and is often associated with WmII (yellow in Figure 7d). Grt$_{IId}$ is localised between the large garnet porphyroblast domain and the microlithon, close to WmII, and describe a fine corona rimming Grt$_{IIc}$.

Fe and Ca distribution is quite symmetrical, with Fe content higher within the core of the large garnet porphyroblast (Grt$_I$), and lower in Grt$_{II}$ (Figure 8c). Ca content is low or null in Grt$_I$ and increases in Grt$_{II}$.

Biotite, preserved within ex-sillimanite domains, displays high Ti content especially in biotite cores (Figure 9c). Fe + Mg + Mn content increases from core to rims (Figure 9d). White mica within ex-sillimanite domains (Wm$_I$) shows higher Al contents and lower X$_{Mg}$ with respect to the white mica (Wm$_{II}$) developed within ex-plagioclase microdomain (Figure 10c,d).

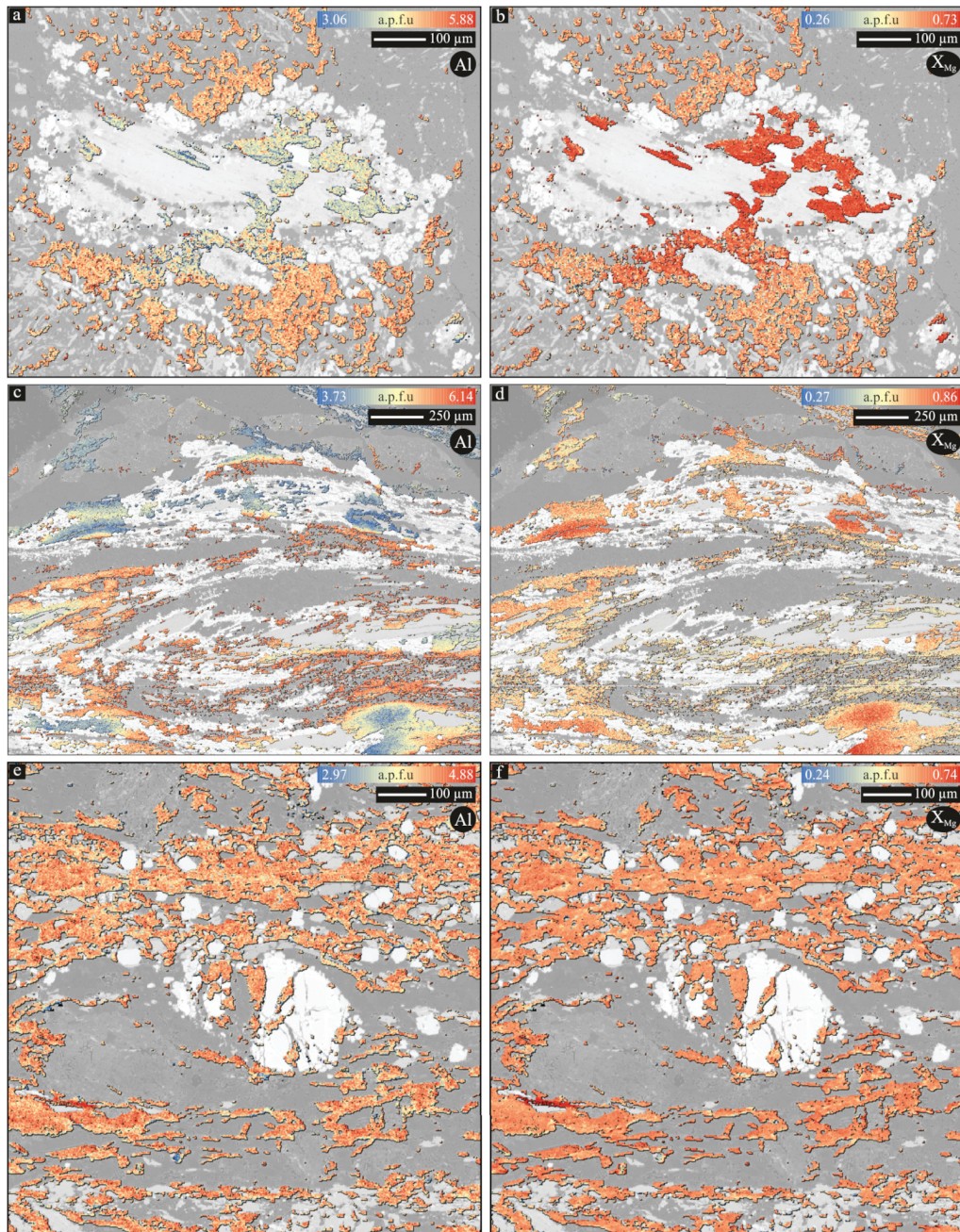

**Figure 10.** Calibrated maps of white mica: (**a**,**b**) calibrated maps of Al and $X_{Mg}$ of LdV white mica; (**c**,**d**) calibrated maps of Al and $X_{Mg}$ of MM white mica; (**e**,**f**) calibrated maps of Al and $X_{Mg}$ of RCT white mica.

### 4.3. Rocca Canavese Mylonitic Paragneisses

The RCT orthogneiss are preserved as low strain domain, within a mylonitic zone, and do not display primary texture at outcrop [19]. These mylonites separate the Eclogitic Micaschists Complex from the Rocca Canavese Thrust Sheets. They are constituted by glaucophanites, orthogneisses, paragneisses, metagabbros, and serpentinites (Figure 4). The mylonitic contact runs NE-SW and displays meter scale rootless folds and a premylonite foliation. The mylonite horizon is folded by a fold system with a wavelength of several meters and an axial plane striking NW-SE and dipping 70–80 degrees. Due to the strong mylonitization, pre-Alpine relicts are generally missing, except for garnet porphyroclasts within paragneisses. In these rocks the mylonite foliation is marked by quartz-jadeite rich-microlithons alternate to white mica microfilms (Figure 5e). Garnet

occurs as isolated rounded grains, generally larger than aggregates, aligned to the foliation (Figure 5e).

Figure 6f displays this distribution of main mineral phases, as detected by First Cycle of Q-XRMA. Relative modal abundances of each mineralogical phase are presented in Table 4. Details on microdomain (Figure 7e,f) display $Grt_I$ core rimmed by $Grt_{II}$ or as strain shadow (Figure 6f). $Grt_{II}$ individuals also occur within microfilm domains. White mica also occurs in three generations, $Wm_I$, $Wm_{II}$, and $Wm_{III}$: $Wm_I$ as isolated grains, $Wm_{II}$ is the core of large porphyroblasts; $Wm_{III}$ may form rims or new grains within the microfilm domains. The microfilm domain is also defined by $Gln_I$, $Ep_I$, $Jd_I$, and fine-grained $Wm_{III}$. The latter assemblage marks the S2 foliation (Figure 6e,f). Figure 7 clearly reproduces this mineral assemblage (Figure 7e) but also highlights the various garnet and white mica generations (Figure 7f). It is worth noting as large garnet individuals record the entire spectrum of garnet generations, from $Grt_I$ to $Grt_{III}$, while smaller individuals, within the rock matrix, record $Grt_{II}$ and $Grt_{III}$. $Grt_{II}$ constitutes cores while $Grt_{III}$ rims (Figure 7f). $Grt_I$ displays high Fe and low Ca contents while $Grt_{II}$ and $Grt_{III}$ generations grossly show a decrease in Fe and increase in Ca content (Figure 8e,f). White mica shows a decrease in Al content from $Wm_I$ to $Wm_{III}$ (Figure 10e) whereas $X_{Mg}$ display little variations between the three different generations (Figure 10f).

**Table 4.** Average compositions of minerals from Rocca Canavese obtained via Q-XRMA. All values are calculated on the base of 1528 calibrated pixels for $Grt_{Ia}$; 218 calibrated pixels for $Grt_{Ib}$; 387 calibrated pixels for $Grt_{II}$ and 1299 calibrated pixels for $Grt_{III}$; 1100 calibrated pixels for $Wm_I$; 45,177 calibrated pixels for $Wm_{II}$ and 27,115 calibrated pixels for $Wm_{III}$.

| Mineral | Garnet | | | | White Mica | | |
|---|---|---|---|---|---|---|---|
| Sub-Phase | $Grt_{Ia}$ | $Grt_{Ib}$ | $Grt_{II}$ | $Grt_{III}$ | $Wm_I$ | $Wm_{II}$ | $Wm_{III}$ |
| Al | 1.99 | 1.98 | 1.95 | 1.95 | 4.33 | 4.22 | 4.07 |
| Ca | 0.18 | 0.33 | 1.14 | 1.30 | 0.01 | 0.01 | 0.01 |
| Fe | 2.16 | 2.21 | 1.62 | 1.46 | 0.38 | 0.43 | 0.46 |
| K | 0.00 | 0.00 | 0.00 | 0.00 | 1.79 | 1.80 | 1.81 |
| Mg | 0.42 | 0.18 | 0.09 | 0.06 | 0.57 | 0.56 | 0.56 |
| Mn | 0.26 | 0.29 | 0.17 | 0.19 | 0.01 | 0.00 | 0.01 |
| Na | 0.00 | 0.00 | 0.00 | 0.00 | 0.04 | 0.04 | 0.04 |
| Si | 2.99 | 3.00 | 3.01 | 3.01 | 6.78 | 6.85 | 6.93 |
| Ti | 0.00 | 0.00 | 0.00 | 0.00 | 0.02 | 0.01 | 0.01 |
| Cations | 7.99 | 7.99 | 7.98 | 7.98 | 13.92 | 13.93 | 13.89 |
| Alm | 71.50 | 73.49 | 53.63 | 48.39 | | | |
| Grs | 6.04 | 10.81 | 37.73 | 43.10 | | | |
| Prp | 13.95 | 5.97 | 2.98 | 2.13 | | | |
| Sps | 8.51 | 9.73 | 5.66 | 6.38 | | | |
| $X_{Fe}$ | | | | | 0.40 | 0.43 | 0.45 |

## 5. Petrological and Tectonics Implications

The distribution and composition of the pre-Alpine mineral relicts as well as the new Alpine-formed minerals were successfully depicted using the elemental map data treatment by means of the three operational stages of the Q-XRMA. In the following parts of the work, we will use the obtained results to identify and chemically differentiate obtained local parageneses. These parageneses are often preserved in very limited volumes and are interpreted as representative of the pre-Alpine metamorphic relicts as well as of the reaction products developed during the high pressure Alpine metamorphic stage. Calibrated maps will be used to define the Effective Reactants Volumes (ERV) [98]. This procedure highlights the specific compositional ranges of the isolated parageneses and permitting to define the amount and direction of the reactivity between the different types of reacting boundaries, especially where solid solution phases are in direct contact with each other. Geothermobarometers will be applied to each isolated mineral parageneses on a very statistically meaningful and reliable dataset. The PT estimates will be used to better

constrain the pre-Alpine metamorphic scenario of these Austroalpine key sectors as well as to better understand the influence of the pre-Alpine parageneses and textures on the following Alpine tectonic evolution.

### 5.1. Mineral Phase Compositions and Parageneses

Samples from the three localities preserve specific mineral relics in various domains, which differently controlled the local reactivity evolution of the new Alpine formed parageneses. Within these domains, different chemical compositions were isolated and subdivided per specific compositional range as averagely reported in Tables 2–4. These compositions were then interpreted as representative of a specific recognized mineral metamorphic stage averaging the results of the calibrations previously obtained (Figures 10 and 11). This permitted to obtain a more meaningful and reliable dataset, since the calibration permitted to observe the total mineral compositional reactivity, hardly to achieve just through the single spot EPMA analyses.

Generally, the preserved mineral phases are garnet, white mica, and biotite (Figures 9 and 11), while the new Alpine minerals are garnet and white micas. LdV preserves igneous biotite and white mica and new Alpine garnet and white micas. MM preserves metamorphic garnet, biotite, and white mica, and new Alpine garnet and white micas during the Alpine stages. Finally, RCT preserves metamorphic garnet and white mica and forms again new Alpine garnet and white micas.

More specifically, it is possible to observe as pre-Alpine LdV mineral relicts are characterised by a relatively constant biotite composition already well defined by the EPMA analyses. Two mineral growth stages can be recognised probably due to the chemical re-equilibration developed during the different stages of the pre-Alpine evolution. $Bt_I$ is igneous and it is surrounded by plagioclase and or K-feldspar domains, now partly or completely replaced by fine-grained aggregates. $X_{Fe}$ content varies from 0.5 to 0.6 while Al from 2.0 to 2.5 a.p.f.u. (Figure 12).

Pre-Alpine igneous white mica shows, instead, a more variable composition recorded in the relict preserved porphyroclasts. Alpine white micas are well developed (see the open red squares in Figure 12) and show Al content increases from $Wm_I$ to $Wm_{III}$. The calibration of this phase permitted to enrich significantly the compositional variability of the reacting white micas, enlarging the compositional range and then obtaining different average composition with respect to those obtained by the average of the EPMA analyses. Even more surprising is the identification of a second type of white mica composition, here ascribed as due to a second stage of the pre-Alpine evolution, totally not recognised by the EPMA analyses (see blu-light points on Figure 12).

Finally, garnet growth of LdV is totally due to the Alpine evolution yet texturally constrained (Figures 5 and 8). Moreover, in this case the calibration permitted to obtain a very important enlarging of the compositional variability of each recognised garnet growth stage, allowing to better define the incremental re-equilibration steps of the Alpine metamorphic effects. In particular, garnet Ia and Ic (Figure 12) were recognised and texturally well-constrained (Figure 8). It is indeed noteworthy as the compositional behaviour of the new micro-garnets around the pre-Alpine biotite is strictly controlled by the mineral direct in contact. Iron composition of the micro-garnet inner corona, directly in contact with biotite, rapidly decreases towards the external part, which is instead in contact with plagioclase. By contrast, an opposite behaviour of calcium is observed, very enriched in the external part with respect to those in contact with biotite, confirming the predisposition of the micro-garnets coronas formation to be very sensitive, for this P-T condition, to be compositionally influenced by the mineral directly contact.

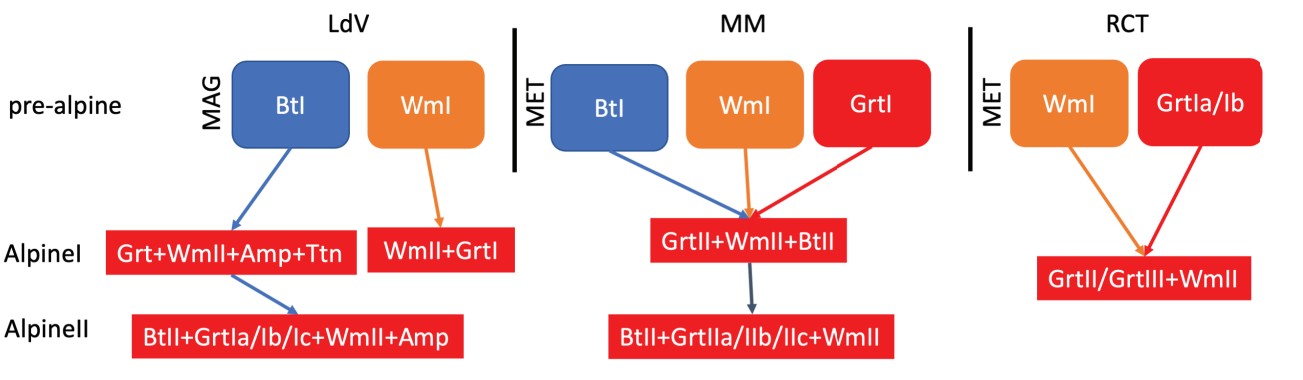

**Figure 11.** Schematisation of pre-Alpine mineralogical assemblages and subsequent Alpine development of mineralogical associations for the three investigated localities.

The MM mineral compositional parageneses see a pre-Alpine garnet with a very constant composition, already well defined by the average composition obtained via EPMA analyses. By contrast, Alpine garnet show a very articulated mineral composition variability, strictly controlled by the mineral directly in contact each other (Figures 8c,d and 12). In this case, the average EPMA analyses badly match the real compositional variability of the new forming Alpine garnet compositions. It is noteworthy, for instance, as the compositions of the $Grt_{IIa}$ and $Grt_{IIc}$ are exclusively constrained by the map calibration after the detailed second cycle sub-phase classification (Figure 7b). MM white micas show a relatively constant composition, passing from the pre-Alpine relicts to the new formed Alpine ones. All the observed composition variability seems to be entirely matched by the EPMA analyses. Finally, biotite highlights a well-defined pre-Alpine compositional variability (Figure 12), already recognised also by the EPMA analysis.

The RCT selected microdomains show a pre-Alpine metamorphic garnets characterised by a classical Mn bell-shaped profile, indicative of prograde clockwise P-T conditions (e.g., [1]). New formed Alpine garnet shows a strong Grs increase (see green points on Figure 12). EPMA analyses depicts relatively well the compositional variability of the observed garnets, even if the calibration permitted to enlarge significantly the data array giving, in turn, a more statistically meaningful result. Finally, RCT white micas show a high celadonite content increase passing from the pre-Alpine relict compositions to the Alpine new forming ones (Figure 12). It is really worth noting as in this case the final celadonite content increase was depicted also by the calibrated pixels not matched by the EPMA analyses.

This synoptic schematisation of Figure 11 put into evidence as the applied method is revealed as a powerful tool to define statistically meaningful parageneses also for very small effective reactants volumes and in very scarce reacting environments. It is finally worth noting that the application of the complete Q-XRMA approach allows the recognition of a new generation of garnet, if compared to previous observations [2,30]. Garnet compositions from the three generations lay on a narrow array, with $Grt_{Ia}$ and $Grt_{Ic}$ showing higher Grs contents and $Grt_{Ib}$ higher Alm content.

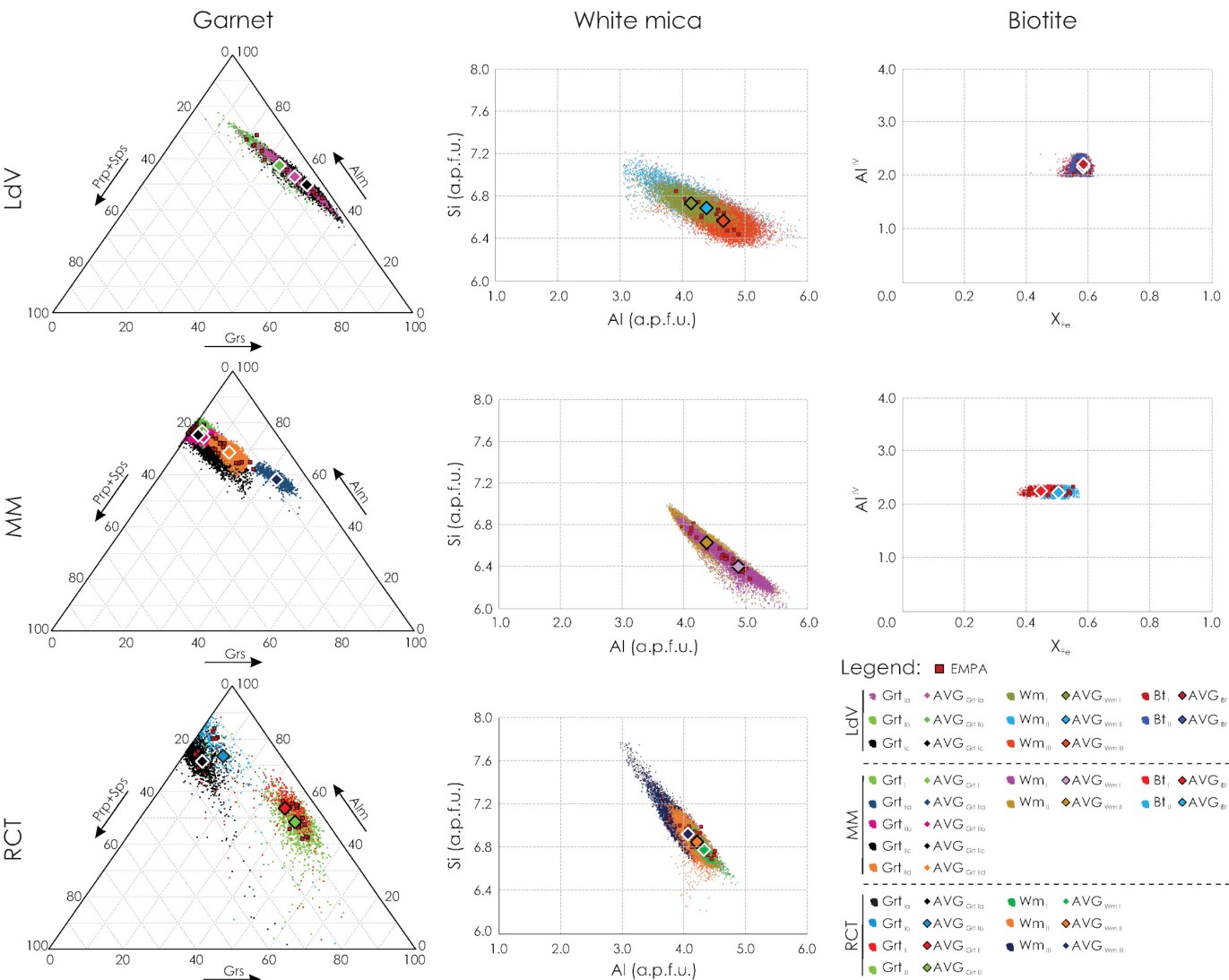

**Figure 12.** Mineral compositions obtained with Q-XRMA for the different generations of garnet, white mica, and biotite from the three investigated localities. AVG refers to the mean compositional values; EMPA = electron microprobe analyses.

## 5.2. Thermobarometry

Thermobarometric estimates were obtained by using the mean compositions reported in Tables 2–4. Temperature is estimated by classical compositional and exchange thermometry: pre-Alpine estimate by Ti in Biotite [99] and Grt-Bt [100–103], Alpine and pre-Alpine temperatures by Grt-Wm [104–107]. Alpine pressure is estimated on phengitic content in white mica [108–110]. Pressures were estimated averaging the available calibrations. It is worth noting that [108] calibration invariably gives the lowest pressure values, providing a minimum pressure estimate. Grt-Wm was tested against different garnet-white mica couples, according to the microstructural and chemical observations, and it was tested at 8–12–16 kbar pressure conditions according to literature values. Table 5 reports calculated values according to the evolutionary scheme reported in Figure 13.

Pre-Alpine magmatic conditions were reconstructed for the LdV metaintrusives. Specifically, temperature estimated using the averaged composition over 8000 pixels of BtI, gives a temperature of 627 °C (Table 5). Pre-Alpine metamorphic conditions were estimated for MM and RCT localities. A temperature above 640 °C was estimated using Ti content in $Bt_I$ and $Bt_{II}$ (Table 5). Lower temperatures were estimated using Grt-Bt exchange at MM (480 ± 60 °C) and at RCT (520 ± 50 °C).

At LdV, temperatures for the Alpine stage were obtained in a range of 420–570 °C applying different calibrations and exchanges (Table 5). Pressure estimates vary depending on the selected microdomain: values of 11–15 kbar were estimated for $Wm_{III}$ and $Wm_{II}$, both included in ex-plagioclase microdomains and associated with albite and quartz (Figure 5). $Wm_{III}$, associated with large $Bt_I$ microdomains, recorded higher pressures, ranging from 14.5 to 16.4 kbar.

At MM, temperatures range between 460 and 600 °C depending on the mineral couple chosen. In particular, $Grt_{IIc}$-$Wm_{II}$ yielded temperature fairly higher than the average Alpine estimates in the Sesia-Lanzo Zone (Figure 13) [33]. $Wm_I$ recorded pressures ranging from 7.1 to 8.8 kbar at temperatures from 450 to 550 °C and ranging from 10 to 11 kbar at 650–700 °C, while $Wm_{II}$ at 450 to 550 °C recorded 12 to 14 kbar.

RCT Alpine estimates were obtained applying the Grt-Wm exchange: the results range from 430 to 480 °C (Table 5). $Wm_I$, $Wm_{II}$ and $Wm_{III}$ Si content yielded pressure estimates of 12–14 kbar within the obtained temperature interval.

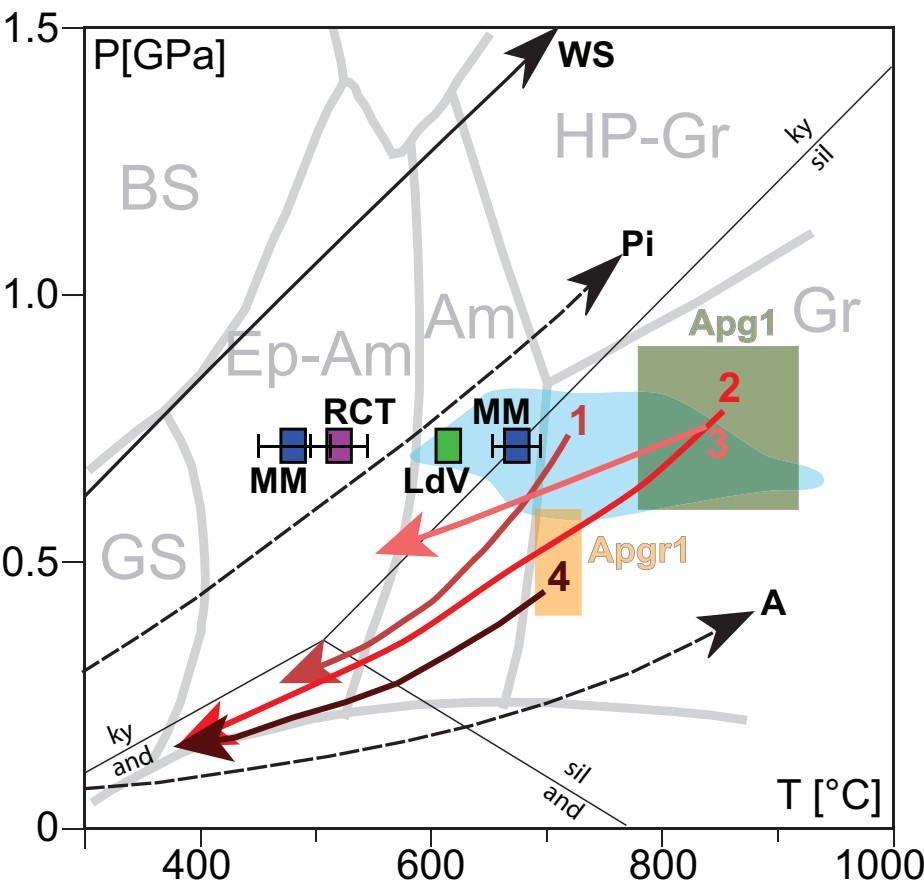

**Figure 13.** Pre-Alpine P-T conditions inferred for Lago della Vecchia (green square, LdV), Monte Mucrone (blue squares, MM), and Rocca Canavese (purple square, RCT) samples compared to the pre-Alpine PT-conditions (cyan shaded area; Table 1) and P-T-t paths from the literature (1: [12]; 2: [20,64]; 3: [64]; 4: [21]). Metagranitoids (Apgr1) and metagabbros (Apg1) emplacement conditions are also reported (Table 1). Metamorphic facies modified after [111–114]. Black arrows represent geothermal gradients traditionally associated to arc regions (A) and plate interior (Pi), warm subduction complexes (WS), and cold subduction complexes (CS), as presented by [115]. Metamorphic facies abbreviations: Am, amphibolite; Am-Ec, amphibole-eclogite; BS, blueschist; Dry-Ec, dry eclogite; Ep-Am, epidote amphibolite; Ep-Ec, epidote-eclogite; Gr, granulite; GS, greenschist; HP-Gr, high pressure-granulite; Lws-Ec, lawsonite-eclogite; PP, prehnite-pumpellyite; Ze, Zeolite.

**Table 5.** Thermobarometric estimates obtained using independent thermometers and barometers and referred to successive stages. (1) Ti in Biotite; (2) Grt-Bt; (3) Grt-Wm; (4) Wm phengitic content.

| Stage | Pre-Alpine Mag | Pre-Alpine Met | | Alpine Met | | |
|---|---|---|---|---|---|---|
| Sample | LdV | MM | RCT | LdV | MM | RCT |
| P(kbar) | | | | 11–15 [@450–550 °C; 4] 14.5–16.4 [@450–550 °C; 4] | 7.1–8.8 [@450–550 °C; 4] 10–11 [@650–700 °C; 4] 12–14 [@450–550 °C; 4] | 12–14 [@400–500 °C; 4] |
| T( °C) | 627 [1] | 690 [1] 642 [1] 480 ± 60 [@6–8 kbar; 2] | 520 ± 50 [@6–8 kbar; 3] | 420–460 [@8–16 kbar; 3] 495 [1] 525 ± 47 [@6–8 kbar; 2] | 530–575 [@8–16 kbar; 3] 563–697 [@8–16 kbar; 3] | 433–470 [@8–14; 3] 441–479 [@8–15; 3] |

### 5.3. Calculation of Mineral Phase Modal Amount

Tables 6 and 7 report the phase percentages as calculated from Q-XRMA outputs, subdivided by microdomains, namely MD1 and MD2. Table 6 displays the results from the Q-XRMA First Cycle, which recognizes main phases but not the different generations. Table 7 represents the result of the Second Cycle, where the mineral generations can be recognized.

We performed the Second Cycle only for the phases of interest (e.g., Grt, Wm, Bt) and the modal content was then calculated. $Grt_I$ is well preserved and in concentrations above the 50%, which are generally higher than those in coronitic or tectonitic domains [7]. Within coronitic domains, as for LdV, pre-Alpine Bt and Wm mica is preserved, with pre-Alpine Bt more abundant (ca. 7%) than the igneous Wm (ca. 3%). Plagioclase is completely replaced by fine-grained aggregates of Wm + Pl.

**Table 6.** Q-XRMA First Cycle—Phases Percentages.

| Zone | μ-Domain | Amp | Ap | Bt | Ep | Grt | Ky | Ox | Pl | Px | Qz | Ttn | Wm | NC | Total |
|---|---|---|---|---|---|---|---|---|---|---|---|---|---|---|---|
| LDV | MD1 | 2.25 | 0.00 | 12.77 | 12.16 | 15.30 | 0.00 | 0.00 | 25.42 | 0.00 | 8.65 | 0.59 | 21.81 | 1.04 | 100.00 |
| MUC | MD1 | 0 | 0.05 | 2.18 | 0.02 | 32.50 | 0.11 | 0.00 | 0.00 | 12.86 | 28.24 | 0.00 | 23.68 | 0.37 | 100.00 |
| | MD2 | 0 | 0.00 | 9.09 | 0.00 | 18.69 | 15.66 | 0.80 | 1.38 | 8.71 | 12.99 | 0.00 | 32.31 | 0.37 | 100.00 |
| RCT | MD1 | 1.38 | 0.05 | 0.00 | 6.08 | 6.44 | 0.00 | 0.00 | 0.00 | 21.97 | 22.79 | 1.29 | 39.99 | 0.01 | 100.00 |

**Table 7.** Q-XRMA Second Cycle—Sub-Phases Percentages.

| Zone | μ-Domain | Garnet | | | | | Biotite | | White Mica | | |
|---|---|---|---|---|---|---|---|---|---|---|---|
| LDV | MD1 | $Grt_{Ia}$ 6.39 | $Grt_{Ib}$ 14.91 | $Grt_{Ic}$ 78.70 | | | $Bt_I$ 54.14 | $Bt_{II}$ 45.86 | $Wm_I$ 16.90 | $Wm_{II}$ 29.15 | $Wm_{III}$ 53.95 |
| MUC | MD1 | $Grt_I$ 66.13 | $Grt_{IIa}$ 0.92 | $Grt_{IIb}$ 10.49 | $Grt_{IIc}$ 17.44 | $Grt_{IId}$ 5.02 | $Bt_I$ 53.40 | $Bt_{II}$ 46.60 | $Wm_I$ 76.16 | $Wm_{II}$ 23.84 | |
| | MD2 | 0.00 | 0.00 | 14.37 | 84.72 | 0.91 | 75.29 | 24.71 | 54.31 | 45.69 | |
| RCT | MD1 | $Grt_{Ia}$ 44.52 | $Grt_{Ib}$ 6.35 | $Grt_{II}$ 11.28 | $Grt_{III}$ 37.85 | | | | $Wm_I$ 1.50 | $Wm_{II}$ 61.56 | $Wm_{III}$ 36.95 |

### 5.4. Tectonic Implications

In the Sesia-Lanzo Zone, the evolution from granulite to eclogite stage is characterised by a post-granulite stage before the onset of the Alpine convergence [12,20,64]. This stage is generally not reported in papers describing relict pre-Alpine stages (Figure 13 and references included). This can be explained as (i) lack of tracers of this stage, (ii) incomplete development of this stage, (iii) complete re-equilibration under the Alpine metamorphic conditions. In the three cases considered, only the LdV shows evidence of a HT metamorphic re-equilibration stage after magma emplacement and predating the Alpine blueschists-eclogite facies conditions.

### 5.5. Pre-Alpine Evolution

Figure 13 summarizes the pre-Alpine P-T paths for the Sesia-Lanzo Zone, including those reconstructed in this work. The metamorphic temperature estimates cluster at 600–700 °C and 450–550 °C. Lower temperatures are generally referred to a later generation of biotite and can be due to a retrograde metamorphic re-equilibration, not fully recorded.

Pressures were taken from the literature and range between 6 and 8 kbar. The PT trajectories reconstructed with the thermobarometrical estimates lay between Arc regions (A) and Plate Interior (Pi) geothermal gradients. These results confirm a high geothermal gradient during Permian times generating the destabilization of the lower and middle crust and large volumes of magma sourcing from the crust or mantle [27]. They also agree with the record of other remnants of the pre-Alpine continental crust, which is preserved in the western Austroalpine domain, e.g., IIDK and Valpelline Series (VP) [32,116]. IIDK and VP recorded a granulite facies stage occurred at Permian time, which is associated with penetrative fabrics and folds. In the IIDK and VP, pegmatite dykes are the only intrusive rocks associated with the Permian HT migmatitic metamorphism. Differently, in the EMC, the granulite metapelites are associated with large intrusive bodies of Permian age (e.g., Monte Mucrone, Monte Mars, Lago della Vecchia), thus representing a magma-rich crust. However, in EMC and RCTU, the recognition of intrusive bodies could have been limited by the intense Alpine metamorphic and mechanical re-equilibration that produced pervasive mylonite foliations and folds at km-scale, all marked by eclogite facies minerals; thus, in these units, Permian intrusive bodies could have been more abundant and larger.

### 5.6. Alpine Evolution

Although the Alpine evolution was not the main goal of this work, Alpine pressure and temperature conditions were successfully calculated from the extracted quantitative mineral compositions. They give temperature in a range from 420 to 550 °C for pressure between 12 and 16 kbar. MM also displays few higher temperature results (550–600 °C), commonly considered above the thermal limit reached by the Sesia-Lanzo Zone during the eclogitic metamorphism. Recently similar high temperatures were obtained [64] for the internal part of the Sesia-Lanzo Zone.

### 5.7. Rock Memory during Tectonic Cycles

The concept of rock matrix, introduced and described mathematically by [6], suggests the number of rocks that theoretically can generate from one protolith, as a function of a given number of deformation and metamorphic stages experienced. However, the possible number of rocks mathematically predicted is not necessarily equal to the number of rocks that are actually seen in the field [6]; for example, in the Sesia-Lanzo Zone, only 12–14 of the theoretical 1024 possible rock types outcrop, most likely due to the strong metamorphic re-equilibration under eclogite facies conditions [6].

Even under such strong re-equilibration, the methodology applied allowed us to detect and study pre-Alpine relicts that are more diffused than previously thought. Thus, the present work confirms that most of the Sesia-Lanzo Zone protoliths, within the EMC, IIDK and RCTU underwent high-geothermal gradient during the Permian. In addition, the pristine chemistry of some pre-Alpine relicts, made available by the incomplete re-equilibration during the Alpine evolution, offer reliable foundation for the estimation of pressure and temperature conditions. It is worth noting that the abundance of such relicts (e.g., igneous, or metamorphic biotite) is generally below 10% relative to the entire volume (Table 7). This estimate agrees with kilometre-scale 3D modelling, where relicts commonly occur in similar percentages [7,8].

## 6. Conclusions

Pre-Alpine metamorphic relicts from meta-acidic rocks from three sites of the Sesia-Lanzo eclogitic continental crust were here investigated to quantitatively extrapolate the reacting zones developed along the boundaries of the pre-Alpine crystal parageneses and

the new Alpine ones. The quantitative data treatment of the X-ray images enabled us to sequentially obtain a former phase, followed by a later sub-phase classification, which permitted to quantitatively investigate the effective bulk rock chemistry of very local paragenetic equilibria.

Electron microprobe analysis of the main phase recognized and sub-phase segmented microdomains were used to calibrate each pixel of the original WDS X-ray maps, permitting to meaningfully extrapolate the specific compositional ranges of both the relict paragenetic equilibria and the new-formed ones. Yielded pre-Alpine and Alpine mineral parageneses were then used to apply different geothermobarometers.

The method applied in this work permitted to highlight how, also from low-reacting systems involved in deep metamorphic re-equilibration stages, it is possible to isolate metamorphic relicts, which are inputs to reliable PT constraints on a very meaningful statistical dataset. Even in a same, or similar, rock type, heterogeneities may develop during metamorphism that may lead to different metamorphic parageneses during a subsequent metamorphic stage.

The results obtained for the selected meta-acidic rocks permitted to extend to the internal western Austroalpine domain, the pre-Alpine amphibolite-granulite facies stage occurred at Permian time, before being re-equilibrated in eclogite-facies conditions during the subducting stage.

**Author Contributions:** Conceptualization, data acquisition and analysis, M.Z., L.C., M.R., G.O., R.V. and D.Z.; writing and reviewing, M.Z., L.C., M.R., G.O., R.V. and D.Z.; drafting, writing and review M.Z., L.C., M.R., G.O., R.V. and D.Z. All authors have read and agreed to the published version of the manuscript.

**Funding:** M.Z., M.R., L.C. and D.Z.—MIUR project "Dipartimenti di Eccellenza 2017-Le geoscienze per la societa´: risorse e loro evoluzione". R.V. and G.O.—University of Catania (PIAno di inCEntivi per la RIcerca di Ateneo 2020/2022-Pia.Ce.Ri), Grant Number: 22722132153, within the project: "Combined geomatic and petromatic applications: The new frontier of geoscience investigations from field- to micro-scale-(GeoPetroMat)".

**Institutional Review Board Statement:** Not applicable.

**Informed Consent Statement:** Not applicable.

**Acknowledgments:** Four anonymous reviewers are gratefully acknowledged for their highly constructive criticism of the text. We thank Iole Spalla for fruitful discussions around the Sesia-Lanzo Zone, during many years. Andrea Risplendente is also thanked for the valuable support with XR-MAPS and EMPA analyses.

**Conflicts of Interest:** The authors declare no conflict of interest. The funders had no role in the design of the study; in the collection, analyses, or interpretation of data; in the writing of the manuscript, or in the decision to publish the results.

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
