# Peer review of "Quantitative X-ray Maps Analaysis of Composition and Microstructure of Permian High-Temperature Relicts in Acidic Rocks from the Sesia-Lanzo Zone Eclogitic Continental Crust, Western Alps"

_minerals, doi:10.3390/min11121421_

Round 1
Reviewer 1 Report
The paper “Composition and microstructure of Permian high-temperature relicts in acidic rocks from Lago della Vecchia, Monte Mucrone, and Rocca Canavese, Sesia-Lanzo Zone eclogitic continental crust, Western Alps” by Michele Zucali et al. provides new and interesting data about the Sesia-Lanzo zone in the western Alps using stepwise controlled elemental maps by means of the Quantitative-X ray Maps Analyzer (Q-XRMA). I appreciated very much this methos and the results presented in the paper.
This paper is interesting and well-written. The provided data are original and accurate, and are presented in very clear way. The interpretations as well as the conclusions are coherent with the data reported in the paper. I think that this paper will represent a fundamental contribution that substantially improves the knowledge of complicated setting of the Sesia-Lanzo Zone.
On the whole I recommend the publication on MINERALS of this paper with minor modifications, that are listed below.
- I suggest to the authors to better define in the Introduction paragraph the sampling strategy, i.e., why has been studied the three complexes and why the three zones have been selected. Only few sentences are necessary.
- I prefer a more classical figure 1 with the geology of the Sesia-Lanzo Zone in the framework of the Western Alps. So, I suggest to change the FIG. 2a in Fig. 1 and in the meantime add more graphical information (i.e., a general cross section) for the readers not familiar with alpine geology. This is just a suggestion.
- In the caption of Fig. 4 localities is repeated twice.
- A very important conclusion of this paper is “The obtained results for the selected meta-acidic rocks permitted to extend for the internal western Austroalpine domain, the pre-Alpine amphibolite-granulite facies stage at Permian time, before to be re-equilibrated in eclogite-facies conditions during the subducting stage”. This conclusion is very interesting but requires 1) a more information in the Geological Setting paragraph about the pre-alpine geology of the Sesia-Lanzo Zone and 2) more discussion about the implications of these results.
Author Response
Dear Reviewer, thank you for you work and suggestions.
We made all the corrections and improvements proposed by the first reviewer
The details are reported in the track-changes file.
Regards
Reviewer 2 Report
This paper analised samples from three main different localities of the Sesia-Lanzo Zone (SLZ) applying an innovative tool for the x-ray maps analisis. The results provide new constraints on pre-alpine and alpine P-T evolution of the SLZ. The paper is well written, the data are well presented and the (poor) conclusions are supported by data, Figures are necessary but they need some improvements (see punctual comments on the attached pdf).
I suggest to accept the paper for publication after minor revision. I have few main comments that should be addressed before publication in Minerals.
1) T range obtained for the pre-Alpine stage is quite large (> 200°C). Such range of T values could be related to protolith variations/strain partitioning/effect of local bulk composition/local overpressure effects? The different study areas are far enough away from each other to justify different pre-Alpine metamorphic evolution inside SLZ?
In addition, your pre-alpine P-T estimations are quite different from the literature ones you reported in fig.12 (there is a gap of almost 100-150°C!).
Please discuss these two points a bit more. (comments on Fig.11 may help!)
2) Figures: there are some problems, especially in Fig.1 and Fig.2. I suggest (in addition to punctual comments) to switch Fig. 1 and Fig. 2, showing the Western Alps and the location of the study areas (with detailed maps) first, and then the three maps with the location of the samples (and all the interesting information for each dot).
Figure 11 should be improved with the complete unraveled evolution (including the igneous phase of LdV samples). A diagram that links calculated thermobarometric estimations with each igneous/metamorphic mineral assemblage may help. This will be useful for the reader and will enrich the discussions, helping to locate more precisely the P-T estimations in precise moments of the evolution. This latter point is not mandatory.
Other figures need just some punctual improvements (see comments on pdf).

Author Response
Dear Reviewer,
thank you for your work and suggestions.
We addressed all the requests made by adding the information and making variations to figures, tables and text.
Line 87; the citation is not needed because this is just the name of the western Periadriatic segment and also it is common knowledge.
The details are reported in the track-changes file.
Regards
Reviewer 3 Report
In article was intended to explore and enhance the parageneses were interpreted as related to the pre-Alpine metamorphic or magmatic stages as well as to local Alpine re-equilibrations. On the basis of electron microprobe analysis specific compositional ranges were defined in micro-domains of the relict and new paragenetic equilibria. In this way calibrated compositional maps were obtained and used to contour different types of reacting boundaries between adjacent solid solution phases.
The text was prepared very carefully but in my opinion in a lot place must was improved. I have a some questions of the presented process, too. Below the suggested my corrections and my questions was presented.
Corrections:
- Language: The English language in the paper should be checked by a professional
- Please reduce the title by using some innovative keywords for the title
- The author use different abbreviations at different places, which confused the reader, Please provide the list of the abbreviation, please use at the start.
- The introduction needs to be more emphasized on the research work with a detailed explanation of the whole process considering past, present, and future scope.
- The introduction part it is too short for this research item.
- Section descriptions should be written as a separate paragraph at the end of the Introduction, by including the sources of material and findings of this work.
- Research gaps should be highlighted more clearly and future applications of this study should be added. Please make the link between the advanced images used in this paper with the petrogenetic characterized.
- Figures are generated through different software or taken from the different means, please use the same tool for producing the figures. This will improve the presentation of the paper and also ease of reading for the reader.
- The conclusion part is a little bit prolix and should be improved. It seems to be a sum of results. Also, limitations, future scope, and recommendations of this study are suggested to be written as a separate section. What was the research significance of this study?
- In some parts of the manuscript, authors discussed the results in light of the literature prior to presenting their results, which is distracting.
- Please kindly make revisions to the language of the paper presentation. There are still some minor typos and grammatical errors.
Summary:
Overall, the research method and analysis of results are well-documented and make sense. The authors presented modest introduction literature overview, too. The subject and the methodology applied seem to be justified and deserve publication but after major reviewers. The topic is very important for the petrogenesis from Lago della Vecchia, Monte Mucrone,
and Rocca Canavese, Sesia-Lanzo Zone eclogitic continental
crust, Western Alps.
Author Response
Dear Reviewer,
thank you for your work and suggestions.
We followed most of the suggestions posed. We only skipped the request to add a specific paragraph dedicated to limitations, future scope, and recommendations. We think this kind of paragraph should exist for a methodological paper not this kind of contribution. But we added a few words to address those points.
The details are reported in the track-changes file.